# From discrete-time policies to continuous-time diffusion samplers: Asymptotic equivalences and faster training

**Julius Berner**[*]
*California Institute of Technology*
*NVIDIA*
jberner@nvidia.com

**Lorenz Richter**[*]
*Zuse Institute Berlin*
*dida Datenschmiede GmbH*
richter@zib.de

**Marcin Sendera**[*]
*Jagiellonian University*
*Mila, Université de Montréal*
marcin.sendera@uj.edu.pl

**Jarrid Rector-Brooks**
*Mila, Université de Montréal*
*Dreamfold*
*California Institute of Technology*
jarrid.rector-brooks@mila.quebec

**Nikolay Malkin**
*University of Edinburgh*
*CIFAR Fellow, Learning in Machines and Brains*
nmalkin@ed.ac.uk

**Reviewed on OpenReview:** *https://openreview.net/forum?id=xLE3xJUuDO*

## Abstract

We study the problem of training neural stochastic differential equations, or diffusion models, to sample from a Boltzmann distribution without access to target samples. Existing methods for training such models enforce time-reversal of the generative and noising processes, using either differentiable simulation or off-policy reinforcement learning (RL). We prove equivalences between families of objectives in the limit of infinitesimal discretization steps, linking entropic RL methods (GFlowNets) with continuous-time objects (partial differential equations and path space measures). We further show that an appropriate choice of coarse time discretization during training allows greatly improved sample efficiency and the use of time-local objectives, achieving competitive performance on standard sampling benchmarks with reduced computational cost.

## 1 Introduction

We consider the problem of sampling from a distribution on $\mathbb{R}^d$ with density $p_{\text{target}}$, which is described by an unnormalized energy model $p_{\text{target}}(x) = \exp(-\mathcal{E}(x))/Z$ with $Z = \int_{\mathbb{R}^d} \exp(-\mathcal{E}(x))\,\mathrm{d}x$. We have access to $\mathcal{E}$ but not to the normalizing constant $Z$ or to samples from $p_{\text{target}}$. This problem is ubiquitous in Bayesian statistics and machine learning and has been an object of study for decades, with Monte Carlo methods (Duane et al., 1987; Roberts & Tweedie, 1996; Hoffman et al., 2014; Leimkuhler et al., 2014; Lemos et al., 2023) recently being complemented by deep generative models (Albergo et al., 2019; Noé et al., 2019; Gabrié et al., 2021; Midgley et al., 2023; Akhound-Sadegh et al., 2024).

Building upon the success of diffusion models in data-driven generative modeling (Sohl-Dickstein et al., 2015; Ho et al., 2020; Dhariwal & Nichol, 2021; Rombach et al., 2021, *inter alia*), recent work (*e.g.*, Zhang & Chen, 2022; Berner et al., 2022; Vargas et al., 2023; Richter & Berner, 2024; Vargas et al., 2024; Sendera et al., 2024) has proposed solutions to this problem that model generation as the reverse of a diffusion (noising) process in discrete or continuous time (Fig. 1). Thus $p_{\text{target}}$ is modeled by gradually transporting samples, by a sequence of stochastic transitions, from a simple prior distribution $p_{\text{prior}}$ (*e.g.*, a Gaussian) to the target distribution. When a dataset of samples from $p_{\text{target}}$ is given, diffusion models are trained using a score matching objective equivalent to a variational bound on data log-likelihood (Song et al., 2021a). The problem is more challenging when we have no samples but can only query the energy function, as training methods

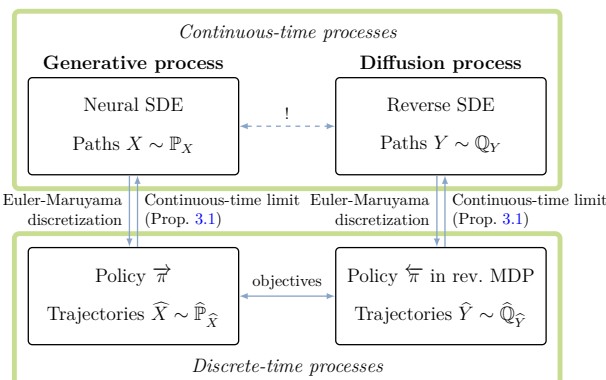

Figure 1: The problem of making continuous-time forward and reverse processes determine the same path space measure is approximated by matching distributions over discrete-time trajectories.

necessarily involve simulation of the generative process. (We survey additional related work in §A.)

In continuous time, we assume the generative process takes the form of a stochastic differential equation (SDE) (with initial condition $p_{\text{prior}}$ and diffusion coefficient $\sigma$):

$$\mathrm{d}X_t = \overrightarrow{\mu}(X_t, t)\,\mathrm{d}t + \sigma(t)\,\mathrm{d}W_t, \quad X_0 \sim p_{\text{prior}}. \tag{1}$$

When the drift $\mu$ is given by a parametric model, such as a neural network, (1) is called a *neural SDE* (Tzen & Raginsky, 2019; Kidger et al., 2021a; Song et al., 2021b). The goal is to fit the parameters so as to make the distribution of $X_1$ induced by the initial conditions and the SDE (1) close to $p_{\text{target}}$.

In discrete time, we assume the generative process is described by a Markov chain with transition kernels $\overrightarrow{\pi}_n(\widehat{X}_{n+1} \mid \widehat{X}_n)$, $n = 0, \ldots, N-1$, and initial distribution $\widehat{X}_0 \sim p_{\text{prior}}$. The goal is to learn the transition probabilities $\overrightarrow{\pi}_n$ so as to make the distribution of $\widehat{X}_N$ close to $p_{\text{target}}$. This is the setting of stochastic normalizing flows (Hagemann et al., 2023), which are, in turn, a special case of (continuous) generative flow networks (GFlowNets; Bengio et al., 2021; Lahlou et al., 2023).

Training objectives for both the continuous-time and discrete-time processes are typically based on minimization of a bound on the divergence between the distributions over trajectories induced by the generative process and by the target distribution together with the noising process. These objectives may rely on differentiable simulation of the generative process (Li et al., 2020; Kidger et al., 2021b; Zhang & Chen, 2022) or on off-policy reinforcement learning (RL), which optimizes objectives depending on trajectories obtained through exploration (Nüsken & Richter, 2021; Malkin et al., 2023). Objectives may further be classified as global (involving the entire trajectory) or local (involving a single transition). Common objectives and the relationships among them are summarized in Fig. 2.

Any SDE determines a discrete-time policy when using a time discretization, such as the Euler-Maruyama integration scheme; conversely, in the limit of infinitesimal time steps, the discrete-time policy obtained in this way approaches the continuous-time process (Kloeden & Platen, 1992). The question we study in this paper is how the training objectives for continuous-time and discrete-time processes are related in the limit of infinitesimal time steps. We formally connect RL methods to stochastic control and dynamic measure transport with the following theoretical contributions:

(1) We show that global objectives in discrete time converge to objectives that minimize divergences between path space measures induced by the forward and reverse processes in continuous time (Prop. 3.3).

(2) We show that local constraints enforced by GFlowNet training objectives asymptotically approach partial differential equations that govern the time evolution of the marginal densities of the SDE under the generative and noising processes (Prop. 3.4).

These results motivate the hypothesis that an appropriate choice of time discretization during training can allow for greatly improved sample efficiency over the standard practice of training diffusion samplers in a

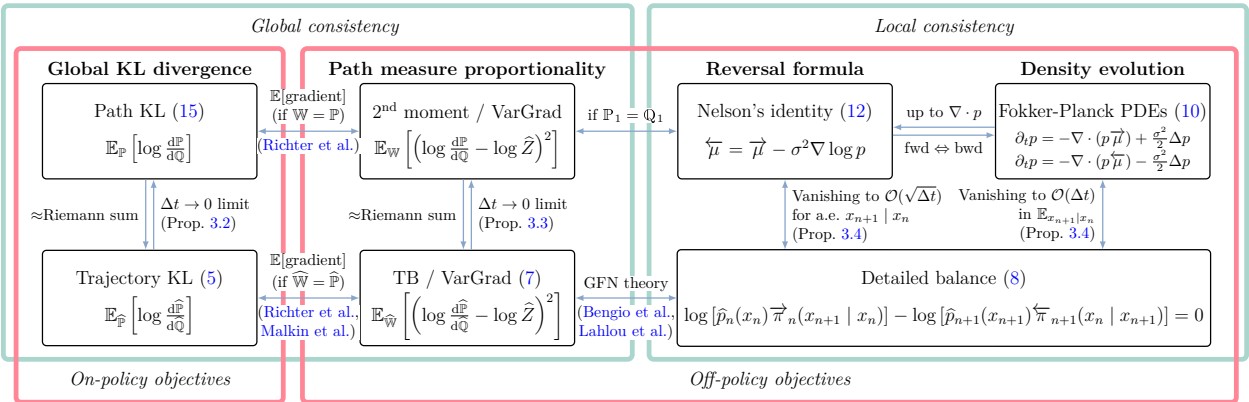

Figure 2: Training objectives for neural SDEs (top row) and their approximations by objectives for discrete-time policies (bottom row). On-policy objectives minimize a divergence by differentiating through SDE integration, while off-policy objectives enforce local or global consistency constraints. Our results explain the behavior of discrete-time objectives as the time discretization becomes finer.

fine, uniform time discretization. Training with shorter trajectories obtained by coarse time discretizations would further allow the use of time-local objectives without the computationally expensive bootstrapping techniques that are necessary when training with long trajectories. Confirming this hypothesis, we make the following empirical contribution:

(3) In experiments on standard sampling benchmarks, we show that training with *nonuniform* time discretizations much coarser than those used for inference achieves similar performance to state-of-the-art methods, at a fraction of the computational cost (Fig. 6).

## 2 Dynamic measure transport in discrete and continuous time

Recall that our goal is to sample from a target distribution $p_{\text{target}} = \frac{1}{Z} \exp(-\mathcal{E}(x))$ given by a continuous energy function $\mathcal{E} \colon \mathbb{R}^d \to \mathbb{R}$. To achieve this goal, we present approaches using discrete-time policies in the framework of Markov decision processes (MDPs) in §2.1 and continuous-time processes in the context of neural SDEs in §2.2. In particular, we will draw similarities between the two approaches and show how time discretizations of neural SDEs give rise to specific policies in MDPs in §2.3. This allows us to rigorously analyze the asymptotic behavior of corresponding distributions and divergences in §3. Note that our general assumptions can be found in §B.1.

Our exposition synthesizes the definitions for MDP policies (Bengio et al., 2023; Lahlou et al., 2023), results on neural SDEs for sampling (Richter & Berner, 2024; Vargas et al., 2024), and PDE perspectives (Máté & Fleuret, 2023; Sun et al., 2024). The results in §3 extend classical results on SDE approximations (see, *e.g.*, Kloeden & Platen (1992)) to objectives for diffusion-based samplers.

### 2.1 Discrete-time setting: Stochastic control policies

A discrete-time Markovian process $\widehat{X}$ with density $\widehat{\mathbb{P}}(\widehat{X})$ – a distribution over $\mathbb{R}^d$-valued variables $\widehat{X}_0, \ldots, \widehat{X}_N$ – can be identified with a policy $\overrightarrow{\pi}$ in the deterministic Markov decision process (MDP) $(\mathcal{S}, \mathcal{A}, T, R)$ depicted in Fig. 3, given by

$$\overrightarrow{\pi}(a \mid \bullet) = \widehat{\mathbb{P}}(\widehat{X}_0 = a) = p_{\text{prior}}(a),$$
$$\overrightarrow{\pi}_n(a \mid (x, t_n)) = \widehat{\mathbb{P}}(\widehat{X}_{n+1} = a \mid \widehat{X}_n = x). \tag{2}$$

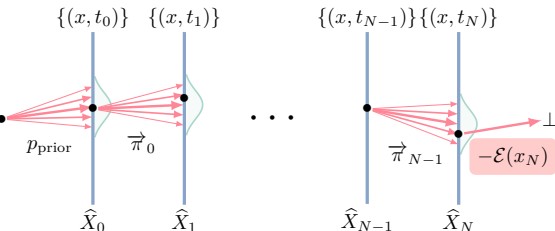

Figure 3: The MDP and policy representing the process $\widehat{\mathbb{P}}$, a distribution over $\widehat{X} = (\widehat{X}_0, \ldots, \widehat{X}_N)$.

We sometimes write $\overrightarrow{\pi}_n(\cdot \mid x)$ for $\overrightarrow{\pi}_n(\cdot \mid (x, t_n))$ for convenience. We relegate formal definitions to §B.2; in short, the states are pairs of space and time coordinates $(x, t_n)$ (together with abstract initial and terminal

states), actions represent steps from $\widehat{X}_n$ to $\widehat{X}_{n+1}$ (taking action $a$ leads to state $(a, t_{n+1})$), and the reward for terminating from a state $(x, t_N)$ is set to $-\mathcal{E}(x)$. The learning problem is to find $\overrightarrow{\pi}$ whose induced distribution over $\widehat{X}_N$ is the Boltzmann distribution of the reward.

This learning problem differs from that of standard reinforcement learning, where one wishes to *maximize* the expected reward with respect to a policy in the MDP. However, it is close to the setting of maximum-entropy reinforcement learning (MaxEnt RL; Ziebart, 2010), in which the objective includes the policy entropy as a regularization term. With such regularization, the optimal policy samples the Boltzmann distribution of the negative reward, which is the target distribution in our setting. This observation is the basis for the connection between control (policy optimization) and inference (sampling) (Levine, 2018). (See Eysenbach & Levine (2022) for the connection to KL-constrained or adversarial policy optimization, Zhang & Chen (2022); Domingo-Enrich et al. (2025) for the related derivation of sampling via stochastic control with a diffusion policy, and Tiapkin et al. (2023); Deleu et al. (2024) for the connections between MaxEnt RL and off-policy objectives for training samplers of discrete sequentially constructed objects. In short, the time-local detailed balance objective (8) minimizes the error in a soft Bellman equation, while trajectory-level objectives (7) minimize a path consistency objective (Nachum et al., 2017).)

**Distributions over trajectories.** The possible trajectories in the MDP starting at $\bullet$ and ending in $\perp$ have the form $\bullet \to (x_{t_0}, t_0) \to \cdots \to (x_{t_N}, t_N) \to \perp$, which we abbreviate to $x_{t_0} \to x_{t_1} \to \cdots \to x_{t_N}$. Following the policy $\overrightarrow{\pi}$ for $N+1$ steps starting at $\bullet$ yields a distribution over trajectories $x_{t_0} \to x_{t_1} \to \cdots \to x_{t_N}$, *i.e.*,

$$\widehat{\mathbb{P}}(\widehat{X}) = \widehat{\mathbb{P}}(\widehat{X}_0) \prod_{n=0}^{N-1} \widehat{\mathbb{P}}(\widehat{X}_{n+1} \mid \widehat{X}_n) = p_{\text{prior}}(\widehat{X}_0) \prod_{n=0}^{N-1} \overrightarrow{\pi}_n(\widehat{X}_{n+1} \mid \widehat{X}_n). \tag{3}$$

The same construction is possible in reverse time: a density $p_{\text{target}}$ over $\widehat{X}_N$ and a policy $\overleftarrow{\pi}$ (analogously to (2) defining transition probabilities from $\widehat{X}_{n+1}$ to $\widehat{X}_n$) on the reverse MDP yields a Markovian distribution over trajectories $\widehat{\mathbb{Q}}$, given analogously to (3) in reverse time. Given a (forward) policy, the reverse policy generating the same distribution over trajectories can be recovered using the marginal state visitation distributions via the detailed balance formula (8).

**Radon-Nikodym derivative and divergences.** The distributions $\widehat{\mathbb{P}}$, $\widehat{\mathbb{Q}}$ determined by a pair of policies $\overrightarrow{\pi}$, $\overleftarrow{\pi}$ and densities $p_{\text{prior}}$, $p_{\text{target}}$ allow us to develop divergences (losses) for learning the parameters of suitable parametric families of policies. Our goal is to make the forward and reverse processes approximately equal by minimizing a divergence between the distributions over their trajectories. The density ratio of these distributions, also known as *Radon-Nikodym derivative*, is given by

$$\frac{d\widehat{\mathbb{P}}}{d\widehat{\mathbb{Q}}}(\widehat{X}) = \frac{\widehat{\mathbb{P}}(\widehat{X})}{\widehat{\mathbb{Q}}(\widehat{X})} = \frac{\widehat{\mathbb{P}}(\widehat{X}_0) \prod_{n=0}^{N-1} \widehat{\mathbb{P}}(\widehat{X}_{n+1} \mid \widehat{X}_n)}{\widehat{\mathbb{Q}}(\widehat{X}_N) \prod_{n=0}^{N-1} \widehat{\mathbb{Q}}(\widehat{X}_n \mid \widehat{X}_{n+1})} = \frac{p_{\text{prior}}(\widehat{X}_0) \prod_{n=0}^{N-1} \overrightarrow{\pi}_n(\widehat{X}_{n+1} \mid \widehat{X}_n)}{p_{\text{target}}(\widehat{X}_N) \prod_{n=0}^{N-1} \overleftarrow{\pi}_{n+1}(\widehat{X}_n \mid \widehat{X}_{n+1})}. \tag{4}$$

Using (4), we can write the *Kullback-Leibler* (KL) divergence as

$$\begin{aligned}
D_{\text{KL}}(\widehat{\mathbb{P}}, \widehat{\mathbb{Q}}) &:= \mathbb{E}_{\widehat{X} \sim \widehat{\mathbb{P}}}\left[\log \frac{d\widehat{\mathbb{P}}}{d\widehat{\mathbb{Q}}}(\widehat{X})\right] \\
&= \mathbb{E}_{\widehat{X} \sim \widehat{\mathbb{P}}}\left[\log p_{\text{prior}}(\widehat{X}_0) + \mathcal{E}(\widehat{X}_N) + \sum_{n=0}^{N-1} \log \frac{\overrightarrow{\pi}_n(\widehat{X}_{n+1} \mid \widehat{X}_n)}{\overleftarrow{\pi}_{n+1}(\widehat{X}_n \mid \widehat{X}_{n+1})}\right] + \log Z.
\end{aligned} \tag{5}$$

Since $\log Z$ is constant, this expression can be minimized via gradient descent on the parameters of the policies, for instance by zeroth-order gradient estimation (REINFORCE; Williams (1992)). If the policies allow for a differentiable reparametrization as a function of noise (*e.g.*, if they are conditionally Gaussian) we can use a deep reparametrization trick, amounting to writing the KL as a function of the noises introduced at each step. In particular, by fitting the parameters of $\overrightarrow{\pi}$ and $\overleftarrow{\pi}$ so that the two processes are approximate time-reversals of one another, we also get an approximate solution to the sampling problem, *i.e.*, $\widehat{X}_N$ is approximately distributed as the target distribution $p_{\text{target}}$. This can be motivated by the *data processing inequality*, which yields that

$$D_{\text{KL}}(\widehat{\mathbb{P}}(\widehat{X}_N), p_{\text{target}}(\widehat{X}_N)) \leq D_{\text{KL}}(\widehat{\mathbb{P}}, \widehat{\mathbb{Q}}). \tag{6}$$

We can also consider other divergences between two measures $\widehat{\mathbb{P}}$ and $\widehat{\mathbb{Q}}$. For instance, the *trajectory balance* (TB, also known as *second-moment*, Malkin et al. (2022); Nüsken & Richter (2021)) and related *log-variance* (LV, also known as *VarGrad*, Richter et al. (2020)) divergences are given by

$$D_{\text{TB}}^{\widehat{\mathbb{W}}}(\widehat{\mathbb{P}}, \widehat{\mathbb{Q}}) = \mathbb{E}_{\widehat{X} \sim \widehat{\mathbb{W}}} \left[ \left( \log \frac{\mathrm{d}\widehat{\mathbb{P}}}{\mathrm{d}\widehat{\mathbb{Q}}}(\widehat{X}) - c \right)^2 \right] \quad \text{and} \quad D_{\text{LV}}^{\widehat{\mathbb{W}}}(\widehat{\mathbb{P}}, \widehat{\mathbb{Q}}) = \text{Var}_{\widehat{X} \sim \widehat{\mathbb{W}}} \left[ \log \frac{\mathrm{d}\widehat{\mathbb{P}}}{\mathrm{d}\widehat{\mathbb{Q}}}(\widehat{X}) \right], \tag{7}$$

where the density ratio inside the square is given by (4) and $\widehat{\mathbb{W}}$ is a reference measure. We are free in the choice of reference measure, which allows for exploration in the optimization task (in RL, this is called *off-policy* training). We note that the second-moment divergence in (7) requires either the choice of a constant $c$ about which the moment is computed, which can be taken to be the normalizing constant $\log Z$ of $p_{\text{target}}$, if it is known, or a learned approximation. The LV divergence coincides with TB when $c$ is taken to be a batch-level estimate of $\log Z$ (see, *e.g.*, Malkin et al. (2023), §2.3)). While estimators of the two divergences in (7) have different variance (which is related to *baselines* in RL), the expectations of their gradients with respect to the policy of $\widehat{\mathbb{P}}$ coincide when $\widehat{\mathbb{W}} = \widehat{\mathbb{P}}$ and are then, in turn, equal to the gradient of the KL divergence (5) (Richter et al., 2020; Malkin et al., 2023). In §2.2, we will see that one can define analogous concepts in continuous time.

**Local divergences.** Instead of looking at entire trajectories, we can as well define divergences locally, *i.e.*, on small parts of the trajectories. To this end, one can define the so-called *detailed balance* (DB) divergence as

$$D_{\text{DB},n}^{\widehat{\mathbb{W}}}(\widehat{\mathbb{P}}, \widehat{\mathbb{Q}}, \widehat{p}) = \mathbb{E}_{\widehat{X} \sim \widehat{\mathbb{W}}} \left[ \log \left( \frac{\widehat{p}_n(\widehat{X}_n) \overrightarrow{\pi}(\widehat{X}_{n+1} \mid \widehat{X}_n)}{\widehat{p}_{n+1}(\widehat{X}_{n+1}) \overleftarrow{\pi}(\widehat{X}_n \mid \widehat{X}_{n+1})} \right)^2 \right], \tag{8}$$

for the time step $n$, where $\widehat{p}_n$ is a learned estimate of the density of $\widehat{X}_n$ for $0 < n < N$, while $\widehat{p}_0 = p_{\text{prior}}$ and $\widehat{p}_N = p_{\text{target}}$ are fixed. Minimizing the DB divergence enforces that the transition kernels $\overrightarrow{\pi}$ and $\overleftarrow{\pi}$ of $\widehat{\mathbb{P}}$ and $\widehat{\mathbb{Q}}$, respectively, are stochastic transport maps between distributions with densities $\widehat{p}_n$ and $\widehat{p}_{n+1}$, for each $n$. If the policies and density estimates jointly minimize (8) to 0 for some full-support reference distribution $\widehat{\mathbb{W}}$ and all $n$, it can be shown that they also minimize the trajectory-level divergences (7); see Bengio et al. (2021) for the discrete case, Lahlou et al. (2023) for the continuous case, Malkin et al. (2023, §C.1) for the connection to nested variational inference (Buchner, 2021), and Deleu & Bengio (2023) for the connection to detailed balance for Markov chains. The divergence used for training may be a (possibly weighted[1]) sum of the DB divergences (8) for $n = 0, \ldots, N - 1$. 'Subtrajectory' interpolations between the global TB objective (7) and the local DB objective (8) exist; see §B.4 and Nüsken & Richter (2023).

**Uniqueness of solutions.** Learning both the generative policy $\overrightarrow{\pi}$ and the time-reversed policy $\overleftarrow{\pi}$ in the general setting as above leads to non-unique solutions. We can achieve uniqueness of the objectives by prescribing $\overleftarrow{\pi}$ (as in diffusion models), adding additional regularizers (as in Schrödinger (half-)bridges), or prescribing the densities $(\widehat{\mathbb{P}}(\widehat{X}_n))_{n=1}^{N-1}$ and imposing constraints on the policies (as in annealing schemes); see Blessing et al. (2024, Tables 6 & 7) and Sun et al. (2024, Table 1).

## 2.2 Continuous-time setting: Neural SDEs

We consider neural stochastic differential equations (neural SDEs) with isotropic additive noise, *i.e.*, families of stochastic processes $X = (X_t)_{t \in [0,1]}$ given as solutions of SDEs of the form

$$\mathrm{d}X_t = \overrightarrow{\mu}(X_t, t) \, \mathrm{d}t + \sigma(t) \, \mathrm{d}W_t, \qquad X_0 \sim p_{\text{prior}}, \tag{9}$$

where $\overrightarrow{\mu} \colon \mathbb{R}^d \times [0,1] \to \mathbb{R}^d$ is the *drift* (or *control function*), parametrized by a neural network[2]; $\sigma \colon [0,1] \to \mathbb{R}_{>0}$ is the *diffusion rate*, which in this paper is assumed to be fixed (more generally, it could be a $d \times d$ matrix that depends also on $X_t$); and $W_t$ is a standard $d$-dimensional Brownian motion. Using a time discretization, the drift $\overrightarrow{\mu}$, together with the noise given by the diffusion rate and the Brownian motion, can be connected to a policy $\overrightarrow{\pi}$ of a MDP, which can be sampled to approximately simulate the process $X$ (see §2.3).

---

[1]Our result Prop. 3.4 suggests a weighting of $\frac{1}{N\Delta t_n}$, in the notation of §2.3, but our experiments showed no significant difference between such a weighting and a uniform one.

[2]For notational convenience, we do not make the dependence of $X$ on the neural network parameters explicit.

**Distributions over trajectories.** Similar to the previous section, we can define a measure on the trajectories of the process $X$. Since the trajectories $t \mapsto X_t$ are almost surely continuous, the distribution (also known as *law* or *push-forward*) of the process $X$ defines a *path space measure* $\mathbb{P}$, which is a measure on the space $C([0,1], \mathbb{R}^d)$ of continuous functions, representing the distribution of trajectories of $X$. We will show in §2.3 that such a path measure can be interpreted as the limit of distributions over discrete-time trajectories as in (3) when the step-sizes $t_{n+1} - t_n$ tend to zero.

We can also define the time marginals $p \colon \mathbb{R}^d \times [0,1] \to \mathbb{R}$, where for each time $t \in [0,1]$, $p(\cdot, t)$ gives the density of $X_t$. In measure-theoretic notation, the time marginals are the densities of the pushforwards of the path measure $\mathbb{P}$ by the evaluation maps $X \mapsto X_t$ sending a continuous function (trajectory) to its value at time $t$. Thus, we will also denote the distribution of the time marginals by $\mathbb{P}_t$. The evolution of $p$ is governed by the *Fokker-Planck equation* (FPE), which is the partial differential equation (PDE)

$$\partial_t p = -\nabla \cdot (p \overrightarrow{\mu}) + \frac{\sigma^2}{2} \Delta p, \quad p(\cdot, 0) = p_{\text{prior}}, \tag{10}$$

where $\Delta p$ denotes the Laplacian of $p$. The Fokker-Planck equation generalizes the *continuity equation* for ordinary differential equations, which corresponds to the case $\sigma = 0$. It expresses the conservation of probability mass when particles distributed with density $p(\cdot, t)$ are stochastically transported by the drift $\overrightarrow{\mu}$ and diffused with scale $\sigma$. While such a PDE perspective is only possible in continuous time, in §3 we derive that certain MDPs satisfy FPEs in the limit of finer time discretizations.

**Reverse process.** As for reverse-time MDPs, we can also define reverse-time SDEs

$$\mathrm{d}X_t = \overleftarrow{\mu}(X_t, t)\, \mathrm{d}t + \sigma(t)\, \mathrm{d}\overleftarrow{W}_t, \qquad X_1 \sim p_{\text{target}}, \tag{11}$$

where $\overleftarrow{W}_t$ is a reverse-time[3] Brownian motion and $\overleftarrow{\mu}$ is a suitable drift, potentially also parametrized by a neural network. This SDE gives rise to another path space measure $\mathbb{Q}$. While in discrete time (§2.1) local reversibility is given by detailed balance (8), in continuous time one can characterize when the path space measure $\mathbb{Q}$ of the reverse-time SDE in (11) coincides with the path space measure $\mathbb{P}$ of the forward SDE in (9) by a local condition known as Nelson's identity (Nelson (1967), also attributed to Anderson (1982)), which states that $\mathbb{Q} = \mathbb{P}$ if and only if

$$\overleftarrow{\mu} = \overrightarrow{\mu} - \sigma^2 \nabla \log p \quad \text{and} \quad \mathbb{Q}_1 = \mathbb{P}_1, \tag{12}$$

where $p$ denotes the densities of $\mathbb{P}$'s time marginals. It can be shown that substituting this expression into the FPE for the backward process recovers the FPE (10) for the forward process, and similarly that the KL divergence, given by (15) below, between the forward and backward processes is zero.

**Radon-Nikodym derivative and divergences.** Since we typically cannot compute the time marginals, we cannot directly use Nelson's identity to solve the sampling problem. However, similar to §2.1, we can establish learning problems to infer the parameters of the neural networks $\overrightarrow{\mu}, \overleftarrow{\mu}$, so that the induced terminal distribution of the forward SDE (9) is close to the target, $\mathbb{P}_1 \approx p_{\text{target}}$, in some suitable measure of divergence.

The tool to define such learning problems is Girsanov's theorem (see, *e.g.*, Särkkä & Solin (2019, Theorem 7.4) and Sottinen & Särkkä (2008, Theorem A.2)), which states the following. Let $\mathbb{P}^{(1)}$ and $\mathbb{P}^{(2)}$ be the path space measures defined by SDEs of the form (9) with drifts $\overrightarrow{\mu}^{(1)}, \overrightarrow{\mu}^{(2)}$. Then, for $\mathbb{P}^{(2)}$-almost every $X \in C([0,1], \mathbb{R}^d)$, the Radon-Nikodym derivative is given by

$$\log \frac{\mathrm{d}\mathbb{P}^{(1)}}{\mathrm{d}\mathbb{P}^{(2)}}(X) = \int_0^1 \frac{\|\overrightarrow{\mu}^{(2)}(X_t, t)\|^2 - \|\overrightarrow{\mu}^{(1)}(X_t, t)\|^2}{2\sigma(t)^2}\, \mathrm{d}t + \int_0^1 \frac{\overrightarrow{\mu}^{(1)}(X_t, t) - \overrightarrow{\mu}^{(2)}(X_t, t)}{\sigma(t)^2} \cdot \mathrm{d}X_t. \tag{13}$$

An intuitive explanation of (13) using a discrete-time approximation can be found in Särkkä & Solin (2019, §7.4) or in the proof of Lemma B.7. The same result holds for reverse-time processes as in (11) with $\mathrm{d}X_t$ replaced by integration against the reverse-time process $\mathrm{d}\overleftarrow{X}_t$. Using a reversible Brownian motion as a

---

[3] We refer to Kunita (2019); Vargas et al. (2024) for details on reverse-time SDEs and backward Itô integration.

reference path measure (see Léonard (2014; 2013)), we can thus express the Radon-Nikodym derivative between the path measures $\mathbb{P}$ and $\mathbb{Q}$ of the forward and reverse-time SDEs in (9) and (11) as Itô integrals:

$$\log \frac{d\mathbb{P}}{d\mathbb{Q}}(X) = \log \frac{p_{\text{prior}}(X_0)}{p_{\text{target}}(X_1)} + \int_0^1 \frac{\|\overleftarrow{\mu}(X_t,t)\|^2 - \|\overrightarrow{\mu}(X_t,t)\|^2}{2\sigma(t)^2}\,dt \\ + \int_0^1 \frac{\overrightarrow{\mu}(X_t,t)}{\sigma(t)^2} \cdot dX_t - \int_0^1 \frac{\overleftarrow{\mu}(X_t,t)}{\sigma(t)^2} \cdot d\overleftarrow{X}_t, \tag{14}$$

see (Vargas et al., 2024, Prop. E.1). A related result was derived by Richter & Berner (2024, Prop. 2.3, Remark A.1) using the conversion formula $\int_0^1 f(X_t,t) \cdot dX_t = \int_0^1 f(X_t,t) \cdot d\overleftarrow{X}_t - \int_0^1 \sigma(t)^2 \nabla \cdot f(X_t,t)\,dt$. By integrating (14) over $X \sim \mathbb{P}$, one can derive that the KL divergence is given by an expression analogous to (5):

$$D_{\text{KL}}(\mathbb{P}, \mathbb{Q}) = \mathbb{E}_{X \sim \mathbb{P}}\left[\log p_{\text{prior}}(X_0) + \mathcal{E}(X_T) + \int_0^1 \left(\frac{\|\overrightarrow{\mu}(X_t,t) - \overleftarrow{\mu}(X_t,t)\|^2}{2\sigma(t)^2} - \nabla \cdot \overleftarrow{\mu}(X_t,t)\right)dt\right] + \log Z. \tag{15}$$

Informally, the derivation uses that in expectation over $X \sim \mathbb{P}$, the integral with respect to $dX_t$ in (14) is the sum of an integral with respect to $\overrightarrow{\mu}(X_t)\,dt$ and a stochastic integral with zero expectation.

The KL divergence can also be interpreted as the cost of a continuous-time stochastic optimal control problem (Dai Pra, 1991; Berner et al., 2022). Some objectives, such as those in Zhang & Chen (2022), optimize the parameters of the drift defining $\mathbb{P}$ by minimizing variants of the KL divergence (15) approximately: by passing to a time discretization of the SDE (§2.3) and expressing the objective as a function of the Gaussian noises introduced at each step of the SDE integration, amounting to a deep reparametrization trick. For suitable integration schemes (Vargas et al., 2023; 2024), the discretized Radon-Nikodym derivative can be written as a density ratio, so that this approach corresponds to optimizing a discrete-time KL as in (5).

Analogously to the discrete-time setting (7), we can also consider the second-moment or log-variance divergences $D_{\text{TB}}^{\mathbb{W}}(\mathbb{P}, \mathbb{Q}) = \mathbb{E}_{X \sim \mathbb{W}}\left[\left(\log \frac{d\mathbb{P}}{d\mathbb{Q}}(X)\right)^2\right]$ and $D_{\text{LV}}^{\mathbb{W}}(\mathbb{P}, \mathbb{Q}) = \text{Var}_{X \sim \mathbb{W}}\left[\log \frac{d\mathbb{P}}{d\mathbb{Q}}(X)\right]$, where $\mathbb{W}$ is a reference path space measure. These divergences were explored by Nüsken & Richter (2021).

**Local time reversal: PDE viewpoint.** The continuous-time perspective also offers to employ the PDE framework for learning the dynamical measure transport. Recall that the density $p$ of the process $X$ defined in (9) fulfills the Fokker-Planck equation (10). One can thus aim to learn $\overrightarrow{\mu}$ so as to make it satisfy the FPE, with the boundary values $p(\cdot, 0) = p_{\text{prior}}$ and $p(\cdot, 1) = p_{\text{target}}$, where $p$ is either prescribed or also learned (as done in Máté & Fleuret (2023)). In Sun et al. (2024, §3.3 and §A.3) it is shown that when using suitable losses on this problem one recovers a loss equivalent to $D_{\text{TB}}$. When choosing the diffusion loss from Nüsken & Richter (2023), one recovers a continuous-time variant of $D_{\text{SubTB}}$ (see §B.4) and thus $D_{\text{DB}}$. In §3, we show that it also works the other way around: we can start with the discrete-time detailed balance divergence and derive PDE constraints in the limit.

### 2.3 From SDEs to discrete-time Euler-Maruyama policies

Simulation of the process $X$ can be achieved by discretizing time and applying a numerical integration scheme, such as the Euler-Maruyama scheme (Maruyama, 1955). Specifically, one fixes a sequence of time points $0 = t_0 < t_1 < \cdots < t_N = 1$ and defines the discrete-time process $\widehat{X} = (\widehat{X}_n)_{n=0}^N$ by

$$\widehat{X}_0 \sim p_{\text{prior}}, \quad \widehat{X}_{n+1} = \widehat{X}_n + \overrightarrow{\mu}(\widehat{X}_n, t_n)\Delta t_n + \sigma(t_n)\sqrt{\Delta t_n}\,\xi_n, \quad \xi_n \sim \mathcal{N}(0, I_d), \tag{16}$$

where $\Delta t_n := t_{n+1} - t_n$. This defines the policy $\overrightarrow{\pi}(a \mid (x, t_n)) = \mathcal{N}(a; x + \overrightarrow{\mu}(x, t_n)\Delta t_n, \sigma(t_n)^2 \Delta t_n)$ on an MDP as in (2). It is clear by comparing (2) and (16) that this distribution exactly coincides with the distribution $\widehat{\mathbb{P}}$ in (3) over sequences $(\widehat{X}_0, \widehat{X}_1, \ldots, \widehat{X}_N)$ of the Euler-Maruyama-discretized process $\widehat{X}$. As we will discuss below, with decreasing mesh size, the marginals $\widehat{\mathbb{P}}(X_n)$ of the $n$-th step of the discretized process converge to the marginals $p(\cdot, t_n)$ of the continuous-time process at time $t_n$. Based on the Central Limit Theorem, such convergence can also be shown for non-Gaussian policies that satisfy suitable consistency conditions (Kloeden & Platen, 1992, §6.2).

Finally, the same discretization is possible for reverse time: a reverse-time process of the form (11) with drift function $\overrightarrow{\mu}$ together with a target density $p_{\text{target}}$ determine a policy $\overleftarrow{\pi}$ on the reverse MDP, corresponding to reverse Euler-Maruyama integration:

$$\widehat{X}_N \sim p_{\text{target}}, \quad \widehat{X}_n = \widehat{X}_{n+1} - \overleftarrow{\mu}(\widehat{X}_{n+1}, t_{n+1})\Delta t_n - \sigma(t_{n+1})\sqrt{\Delta t_n}\,\xi_n, \quad \xi_n \sim \mathcal{N}(0, I_d). \tag{17}$$

However, note that the Euler-Maruyama discretizations of a process and of its reverse-time process defined by (12) do not, in general, coincide. That is, a policy on the reverse MDP can be constructed either by discretizing an SDE to yield a policy on the forward MDP, then reversing it, or by discretizing the reverse SDE to directly obtain a policy on the reverse MDP, possibly with different results. In particular, the Gaussianity of transitions is not preserved under time reversal: the reverse of a discrete-time process with Gaussian increments does not, in general, have Gaussian increments. However, Nelson's identity (12) shows that the two are equivalent in the continuous-time limit.

The discretization allows us to compare the two Radon-Nikodym derivatives: those of the discretizations in (4) and of the continuous-time processes in (14). In particular, in Lemma B.7 we will show that these expressions are equal in the limit.

## 3 Asymptotic convergence

### 3.1 Distributions over trajectories

A standard result shows that the discretized process $\widehat{X}$ converges to the continuous counterpart $X$ as the time discretization becomes finer, *i.e.*, as the maximal step size $\max_{n=0}^{N-1} \Delta t_n$ goes to zero (Maruyama, 1955). The precise statement of convergence requires the processes to be embedded in a common probability space. Let $\iota$ be the mapping from the observation space of $\widehat{X}$ (discrete-time trajectories) to that of $X$ (continuous-time paths) that takes a sequence $\widehat{X}_0, \ldots, \widehat{X}_n$ to the function $f \in C([0,1], \mathbb{R}^d)$ defined by $f(t_n) = \widehat{X}_n$ and linearly interpolating between the $t_n$ (note that $\iota$ implicitly depends on the discretization). We then have convergence of $\iota(\widehat{X})$ to $X$:

**Proposition 3.1** (Convergence of Euler-Maruyama scheme). *As* $\max_{n=0}^{N-1} \Delta t_n \to 0$, $\iota(\widehat{X})$ *converges weakly and strongly to* $X$ *with order* $\gamma = 1$ *and the path measures* $\iota_* \widehat{\mathbb{P}}$ *converge weakly to* $\mathbb{P}$.

We refer the reader to §B.3 for definitions of strong and weak convergence. The result can be found in Kloeden & Platen (1992, §10.2, §10.3) and we refer to Baldi (2017, Corollary 11.1) and Kloeden & Neuenkirch (2007, §2.1) for the convergence of path measures. Generally, the Euler-Maruyama scheme has order of strong convergence $\gamma = 1/2$. However, since we consider *additive* noise, *i.e.*, $\sigma$ not depending on the spatial variable $x$, the *Milstein scheme* reduces to the Euler-Maruyama scheme and we inherit order $\gamma = 1$ as stated in Prop. 3.1 (Kloeden & Platen, 1992).

### 3.2 Radon-Nikodym derivative and divergences

Beyond the convergence of path measures, this section shows – more relevant for practical applications – that commonly used local and global objectives converge their continuous-time counterparts as the time discretization is refined. To this end, we leverage Lemma B.7, which analyzes the convergence of time discretizations of Radon-Nikodym derivatives $\frac{d\mathbb{P}}{d\mathbb{Q}}$ appearing in (14) to their discrete-time analogs $\frac{d\widehat{\mathbb{P}}}{d\widehat{\mathbb{Q}}}$. We note that Vargas et al. (2024, Proposition E.1) shows that, for constant $\sigma$, an Euler-Maruyama discretization of $\frac{d\mathbb{P}}{d\mathbb{Q}}$ can be written as a density ratio as in (4). This also implies that the ratio in the detailed balance divergence in (8) arises from a single-step Euler-Maruyama approximation of the Radon-Nikodym derivative $\frac{d\mathbb{P}}{d\mathbb{Q}}$ on the subinterval $[t_n, t_{n+1}]$. We present proofs of all results in this Section in §B.6.

**Global objectives: Second-moment divergences approach the continuous-time equivalents.** The following key result uses convergence of the Radon-Nikodym derivatives (Lemma B.7):

**Proposition 3.2** (Convergence of functionals). *If $\mathbb{P}, \mathbb{Q}, \mathbb{W}$ are path measures of three forward-time SDEs, the drifts of the SDEs and their derivatives are bounded, and $f \in C^{\infty}(\mathbb{R}, \mathbb{R})$ has at most polynomially growing derivatives, then*

$$\mathbb{E}_{\widehat{X} \sim \widehat{\mathbb{W}}} \left[ f \left( \log \frac{\mathrm{d}\widehat{\mathbb{P}}}{\mathrm{d}\widehat{\mathbb{Q}}}(\widehat{X}) \right) \right] \xrightarrow{\max_n \Delta t_n \to 0} \mathbb{E}_{X \sim \mathbb{W}} \left[ f \left( \log \frac{\mathrm{d}\mathbb{P}}{\mathrm{d}\mathbb{Q}}(X) \right) \right].$$

We now show that the second-moment losses in (7) converge to their continuous-time counterparts.

**Proposition 3.3** (Asymptotic consistency of TB and VarGrad). *Under the assumptions of Prop. 3.2, the divergences $D_{\mathrm{TB}}^{\widehat{\mathbb{W}}}(\widehat{\mathbb{P}}, \widehat{\mathbb{Q}})$ and $D_{\mathrm{LV}}^{\widehat{\mathbb{W}}}(\widehat{\mathbb{P}}, \widehat{\mathbb{Q}})$ converge to $D_{\mathrm{TB}}^{\mathbb{W}}(\mathbb{P}, \mathbb{Q})$ and $D_{\mathrm{LV}}^{\mathbb{W}}(\mathbb{P}, \mathbb{Q})$, respectively.*

The convergence holds for the TB divergence in (7) with respect to any $c$, *i.e.*, $\mathbb{E}_{\widehat{\mathbb{W}}} \left[ \left( \log \frac{\mathrm{d}\widehat{\mathbb{P}}}{\mathrm{d}\widehat{\mathbb{Q}}} - c \right)^2 \right]$, showing that Prop. 3.3 continues to hold if one uses a learned estimate of the log-partition function $\log Z$ as the scalar $c$ in the TB divergence, as typically done in practice.

**Local objectives: Detailed balance approaches the Fokker-Planck PDE.** Consider a pair of forward and reverse SDEs with drifts $\overrightarrow{\mu}$ and $\overleftarrow{\mu}$, respectively, defining processes $\mathbb{P}$ and $\mathbb{Q}$, and suppose that $\widehat{p} \colon \mathbb{R}^d \times [0,1] \to \mathbb{R}$ is a density estimate with $\widehat{p}(\cdot, 0) = p_{\mathrm{prior}}$ and $\widehat{p}(\cdot, 1) = p_{\mathrm{target}}$.

For $0 \le t < t' \le 1$, consider any time discretization in which $t$ and $t'$ are adjacent time steps ($t_n = t$ and $t_{n+1} = t'$). The discretization defines a pair of policies $\overrightarrow{\pi}, \overleftarrow{\pi}$ corresponding to Euler-Maruyama discretizations of the two SDEs. Let us define the *detailed balance discrepancy*:

$$\Delta_{t \to t'}(x, x') \coloneqq \log \frac{\widehat{p}_n(x) \overrightarrow{\pi}_n(x' \mid x)}{\widehat{p}_{n+1}(x') \overleftarrow{\pi}_{n+1}(x \mid x')}, \tag{18}$$

where we set $\widehat{p}_n(x) = \widehat{p}(x, t_n)$. Recalling the definition (8), we have that

$$D_{\mathrm{DB},n}^{\widehat{\mathbb{W}}}(\widehat{\mathbb{P}}, \widehat{\mathbb{Q}}, \widehat{p}) = \mathbb{E}_{\widehat{Z} \sim \widehat{\mathbb{W}}} \left[ \Delta_{t_n \to t_{n+1}}(\widehat{Z}_n, \widehat{Z}_{n+1})^2 \right]. \tag{19}$$

The following proposition will show that the two SDEs are time reversals of one another if and only if certain asymptotics of the DB discrepancy vanish. It is proved using a technical lemma (Lemma B.8), which shows that the asymptotics of the discrepancy in $h$ are precisely the errors in the satisfaction of Nelson's identity and the Fokker-Planck equation.

**Proposition 3.4** (Asymptotic equality of DB and FPE). *Under the smoothness conditions in Lemma B.8, $\overrightarrow{\mu}, \overleftarrow{\mu}, \widehat{p}$ jointly satisfy Nelson's identity ($\overleftarrow{\mu} = \overrightarrow{\mu} - \sigma^2 \nabla \log \widehat{p}$) at $(x_t, t)$ if and only if*

$$\lim_{h \to 0} \left[ \frac{1}{\sqrt{h}} \Delta_{t \to t+h}(x_t, x_{t+h}) \right] = 0 \quad \text{for almost every } z,$$

*where $x_{t+h} \coloneqq x_t + \overrightarrow{\mu}(x_t, t) h + \sigma(t) \sqrt{h} z$. If in addition*

$$\lim_{h \to 0} \mathbb{E}_{z \sim \mathcal{N}(0, I_d)} \left[ \frac{1}{h} \Delta_{t \to t+h}(x_t, x_{t+h}) \right] = 0,$$

*then the Fokker-Plank equation is satisfied at $(x_t, t)$. If both conditions hold at all $(x_t, t) \in \mathbb{R}^d \times (0,1)$, then $\overrightarrow{\mu}, \overleftarrow{\mu}$ define a pair of time-reversed processes with marginal density $\widehat{p}$.*

In particular, this result shows that if we *impose* a parametrization of $\overrightarrow{\mu}$ and $\overleftarrow{\mu}$ as two vector fields that differ by $\sigma^2 \nabla \log \widehat{p}$, where $\widehat{p}$ is a fixed or learned marginal density estimate, then asymptotic satisfaction of DB implies that the continuous-time forward and backward processes coincide.

**Processes defined by discrete-time reversal.** The generative and diffusion processes play a symmetric role in Prop. 3.4. However, some past work – starting from Zhang & Chen (2022), whose experiment settings we adopt in §4 – has defined $\overleftarrow{\pi}$ as the reversal of the Euler-Maruyama discretization of a forward SDE, rather than as the Euler-Maruyama discretization of a backward SDE, in a case where the former has Gaussian increments. To ensure the applicability of the results to the experiment setting, we need a slight generalization:

**Proposition 3.5** (DB and FPE for Brownian bridges). *The results of Prop. 3.4 hold if $\sigma(t)$ is constant and $\overleftarrow{\pi}$ is the discrete-time reversal of the Euler-Maruyama discretization of the process $\mathrm{d}X_t = \sigma(t)\,\mathrm{d}W_t$, $p_{\mathrm{prior}}(x) = \mathcal{N}(x; 0, \sigma_0 I_d)$.*

The results in this section guarantee that global and local objectives with different discretizations are approximating unique continuous-time objects when $\max_{n=0}^{N-1} \Delta t_n \to 0$. This justifies training and inference of samplers with different discretizations, allowing us to greatly reduce the computational cost of training (see §4). These observations are particularly relevant for diffusion-based samplers which rely on discretization of (partial) trajectories during training. In contrast, for generative modeling, one can use denoising score-matching objectives which can be minimized without any discretization in continuous time.

## 4 Experiments

We evaluate the effect of time discretization on the training of diffusion samplers. We follow the training setting from Sendera et al. (2024) throughout, extending their published code with an implementation of variable time discretization (see §C.1 for details). The following objectives from §2 are considered:

- **Path integral sampler (PIS)** (Zhang & Chen, 2022): The trajectory-level KL divergence (5), which approximates the path space measure KL (15) is minimized via the deep reparametrization trick (*i.e.*, through differentiable simulation of the generative SDE, hence necessarily on-policy).
- **Trajectory balance (TB)** and **VarGrad**: The trajectory-level divergences of the second-moment type (7), optimized either on-policy or using the off-policy local search technique introduced in Sendera et al. (2024). As TB and VarGrad are found to be nearly equivalent in unconditional sampling settings, we consider VarGrad only for *conditional* sampling (see Fig. 9).
- **Detailed balance (DB)**: The time-local detailed balance divergence (8), and its variant **FL-DB**, which places an inductive bias on the log-density estimates – first used by Wu et al. (2020); Máté & Fleuret (2023) and evaluated in the off-policy RL setting by Zhang et al. (2024); Sendera et al. (2024) – that assumes access to the target energy at intermediate time points (see §B.5).

Each objective is additionally studied with and without the **Langevin parametrization (LP)**, a technique introduced by Zhang & Chen (2022) that parametrizes the generative SDE's drift function via the gradient of the target energy. The assumptions made by each objective are summarized in Table 1.

Table 1: Properties of training objectives. Variants with LP also use the intermediate energy gradient.

| Property ↓ Objective → | PIS | TB/VarGrad | DB | FL-DB |
|---|---|---|---|---|
| Time-local | ✗ | ✗ | ✓ | ✓ |
| Off-policy | ✗ | ✓ | ✓ | ✓ |
| Use intermediate energy | ✗ | ✗ | ✗ | ✓ |
| Use energy gradient | ✓ | ✗ | ✗ | ✗ |

The noising process is always fixed to the reverse of a Brownian motion, following Zhang & Chen (2022) and subsequent work. The following densities are targeted:

- Standard targets **25GMM** (2-dimensional mixture of Gaussians), **Funnel** (10-dimensional funnel-shaped distribution), **Manywell** (32-dimensional synthetic energy), and **LGCP** (1600-dimensional log-Gaussian Cox process) as defined in the benchmarking library of Sendera et al. (2024).
- **VAE**: the conditional task of sampling the 20-dimensional latent $z$ of a variational autoencoder trained on MNIST given an input image $x$, with target density $p(z \mid x) \propto p(x \mid z)p(z)$.
- Bayesian logistic regression problems for the **German Credit** and **Breast Cancer** datasets (25- and 31-dimensional, respectively), from the benchmark by Blessing et al. (2024).

We use a well-established primary metric: the ELBO of the target distribution computed using the learned sampler and the true log-partition function, estimated using $N$-step Euler-Maruyama integration. In our notation, the ELBO is $\log \widehat{Z} = \mathbb{E}_{\widehat{X} \sim \widehat{\mathbb{P}}}\big[-\mathcal{E}(\widehat{X}_N) + \log \widehat{\mathbb{Q}}(\widehat{X} \mid \widehat{X}_N) - \log \widehat{\mathbb{P}}(\widehat{X})\big]$ (see (32) for details).

While recent work on diffusion samplers has used a discretization with uniform-length time intervals for both training and sampling, we vary the time discretization. Unless stated otherwise, we evaluate ELBO using $N_{\text{eval}} = 100$ uniform discretization steps. However, during training, we vary the number of time steps $N_{\text{train}}$ and their placement, considering three schemes (see Fig. 4) for partitioning the interval $[0, 1]$:

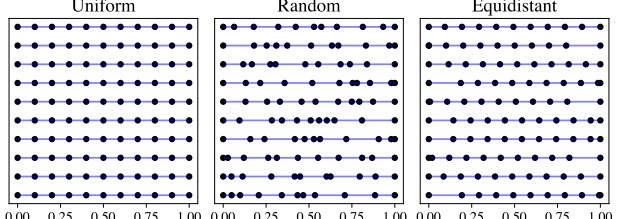

Figure 4: Sampled 10-step discretizations of the unit interval using the three schemes considered.

- **Uniform**: Time steps uniformly spaced: $t_i = \frac{i}{N_{\text{train}}}$ for $i = 0, \ldots, N_{\text{train}}$.
- **Random:** We sample i.i.d. numbers $z_0, \ldots, z_{N_{\text{train}}-1} \sim \mathcal{U}([1, c])$, where $c$ is a constant (we take $c = 10$), and define

$$\Delta t_i = \frac{z_i}{\sum_{j=0}^{N_{\text{train}}-1} z_j}, \quad t_i = \sum_{j=0}^{i-1} \Delta t_j.$$

The interval lengths thus sum to 1, and no two have a ratio of lengths greater than $c$. (We also tested setting the $t_i$ ($0 < i < N_{\text{train}}$) to be i.i.d. samples from $\mathcal{U}([0, 1])$ sorted in increasing order, but this scheme was numerically unstable.)

- **Equidistant:** We sample $t_1 \sim \mathcal{U}([\epsilon, 2/N_{\text{train}} - \epsilon])$, where for us $\epsilon = 10^{-4}$, then set

$$t_i = t_1 + (i - 1)/N_{\text{train}}$$

for $i = 1, \ldots, N_{\text{train}} - 1$. Thus $\Delta t_i = \frac{1}{N_{\text{train}}}$ for all $1 < i < N_{\text{train}} - 1$, *i.e.*, all intervals are of equal length except possibly the first and last.

**Results: Training-time discretization.** In Fig. 5, we show the ELBO gaps on three of the datasets for different training-time discretizations as a function of $N_{\text{train}}$. We observe that, for all objectives, training with **Random** discretization consistently outperforms **Uniform** discretization with a small number of steps, with the two converging as $N_{\text{train}}$ increases to approach $N_{\text{eval}} = 100$. The **Equidistant** discretization performs similarly to **Random** in most cases (see Fig. 10).

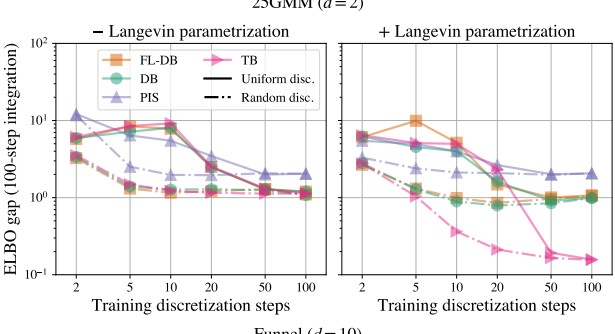
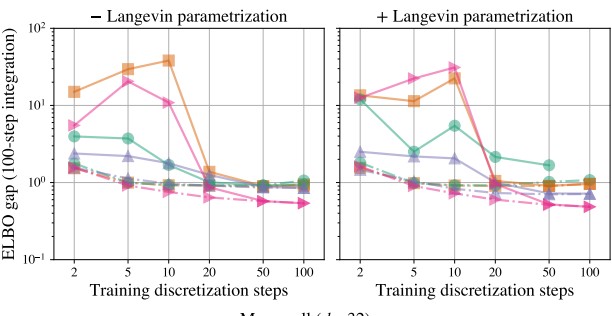
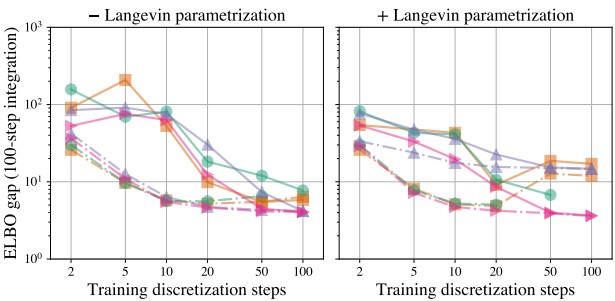

Figure 5: Difference between true $\log Z$ and ELBO as a function of $N_{\text{train}}$, always evaluating with 100-step uniform integration. Additional targets in Fig. 8 and Fig. 9, **Equidistant** results in Fig. 10.

Notably, the time-local objectives (DB, FL-DB) perform similarly to the trajectory-level objectives (TB, PIS) when trained with few steps. However, as $N_{\text{train}}$ increases, the time-local models' performance typically plateaus (on some targets, they diverge with 100 steps). These results suggest that time-local objectives trained with nonuniform discretization and few steps can be a viable alternative to trajectory-level objectives in high-dimensional problems where the memory requirements of training on long trajectories are prohibitive.

**Results: Time efficiency.** The training time per iteration is expected to scale approximately linearly with the trajectory length $N_{\text{train}}$. Fig. 6 (left) confirms this scaling and illustrates the relative cost of different objectives: FL-DB and methods using the Langevin parametrization are the most expensive, as they require stepwise evaluations of the target energy and its gradient, respectively. Fig. 6 (right) shows the ELBO gap plotted against training time, demonstrating that methods with nonuniform discretization achieve a superior trade-off between training time and sampling performance.

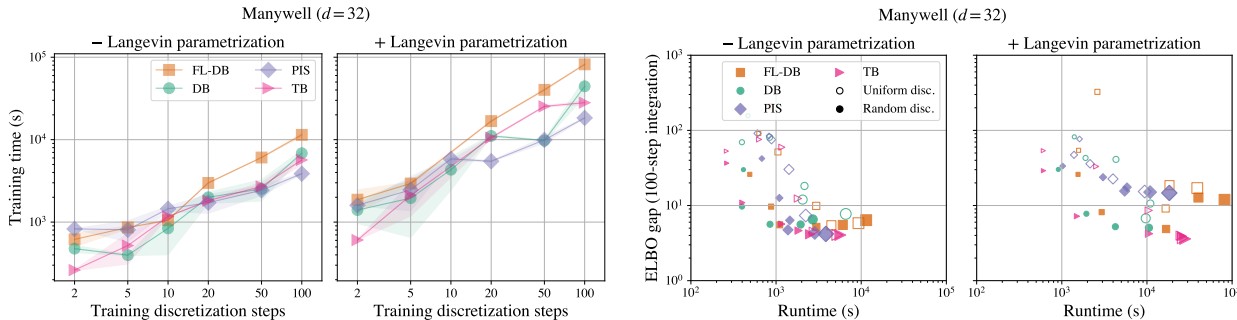

Figure 6: **Left:** Time to train for 25k iterations on **Manywell** as a function of $N_{\text{train}}$, mean and std over 3 runs (note the log-log scale). **Right:** Runtime and ELBO gap, showing that **Random** discretization gives a superior balance of speed and performance. The marker area is proportional to $N_{\text{train}}$. Results for **25GMM** and **Funnel** densities in Fig. 11.

**Results: Inference-time discretization.** To study the effect of *sampling-time* discretization, we train models with $N_{\text{train}} = 10$ steps (using TB with Langevin parametrization) and different placement of time steps, then evaluate with different $N_{\text{eval}} \in \{1, 2, ..., 100\}$. From Fig. 7, we observe that randomized discretization (**Random** or **Equidistant**) during training leads to smooth ELBO curves as a function of $N_{\text{eval}}$, whereas training with **Uniform** discretization gives unstable behavior with periodic features at multiples of $N_{\text{train}}$, which may be due both to the restricted set of inputs $t$ to the model $\overrightarrow{\mu}(x, t)$ during training and to the harmonic timestep embedding in the model architecture. This result is further evidence that nonuniform discretization during training yields more robust samplers that are less sensitive to the choice of $N_{\text{eval}}$.

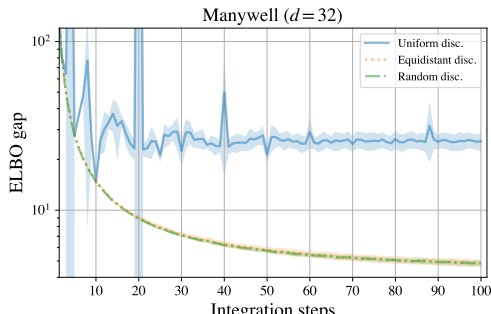

Figure 7: ELBO gaps for models trained with various discretization schemes and $N_{\text{train}} = 10$, then evaluated with various numbers of integration steps $N_{\text{eval}}$. Results on Manywell energy; others shown in Fig. 12.

**Additional results.** Figures complementing those in the main text appear in §D.2 and D.3, while §D.1 contains more metrics and comparisons in tabular form. In particular, we combine the above objectives with the off-policy local search of Sendera et al. (2024) to achieve near-state-of-the-art results with much coarser (nonuniform) time discretizations during training, whereas local search does not help the performance of methods using coarse **Uniform** schemes (Table 2).

## 5 Conclusion

We have shown the convergence of off-policy RL objectives used for the training of diffusion samplers to their continuous-time counterparts. Those are Nelson's identity and the Fokker-Planck equation for stepwise objectives and path space measure divergences for trajectory-level objectives. Our experimental results give a first understanding of good practices for training diffusion samplers in coarse time discretizations. We expect that the increased training efficiency and the ability to use local objectives without expensive energy evaluations are especially beneficial in very high-dimensional problems where trajectory length is a bottleneck, noting that trajectory balance was recently used in fine-tuning of diffusion foundation models for text and images (Venkatraman et al., 2024) and sampling in latent spaces of other large-scale models (Venkatraman et al., 2025). The success of training with coarse, non-uniform steps suggests a path toward such high-dimensional settings.

Future theoretical work could generalize our results to diffusions on general Riemannian manifolds and to non-Markovian continuous-time processes, such as those studied in Daems et al. (2024); Nobis et al. (2024), as well as to discrete-space sampling problems – for which diffusion samplers have recently been developed (Zhu et al., 2025; Guo et al., 2025) – exploiting the unifying connections among diffusion generative processes on general state spaces (Pauline et al., 2025).

## Acknowledgments

We thank Alex Tong, Alexandre Adam, and Pablo Lemos for helpful discussions at early stages of this project.

J.B. acknowledges support from the Wally Baer and Jeri Weiss Postdoctoral Fellowship. The research of L.R. was partially funded by Deutsche Forschungsgemeinschaft (DFG) through the grant CRC 1114 "Scaling Cascades in Complex Systems" (project A05, project number 235221301). The research of M.S. was funded by National Science Centre, Poland, 2022/45/N/ST6/03374.

The research was enabled in part by computational resources provided by the Digital Research Alliance of Canada (https://alliancecan.ca), Mila (https://mila.quebec), and NVIDIA.

## Code

Code is available at https://github.com/GFNOrg/gfn-diffusion/tree/stagger.

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

# A   Additional related work

**Classical sampling methods.** The gold standard for sampling is often considered *Annealed Importance Sampling* (AIS) (Neal, 2001) and its *Sequential Monte Carlo* (SMC) extensions (Chopin, 2002; Del Moral et al., 2006). The former can be viewed as a special case of our discrete-time setting, where, however, the transition kernels are fixed and not learned, thus requiring careful tuning. For the kernels, often a form of *Markov Chain Monte Carlo* (MCMC), such Langevin dynamics and extensions (*e.g.*, ULA, MALA, and HMC) are considered. While they enjoy asymptotic convergence guarantees, they can suffer from slow mixing times, in particular for multimodal targets (Doucet et al., 2009; Kass et al., 1998; Dai et al., 2022). Alternatives are provided by variational methods that reformulate the sampling problem as an optimization problem, where a parametric family of tractable distributions is fitted to the target. This includes mean-field approximations (Wainwright et al., 2008) as well as normalizing flows (Papamakarios et al., 2021). We note that MCMC can also be interpreted as a variational approximation in an extended state space (Salimans et al., 2015).

**Normalizing flows.** There exist various versions of combining (continuous-time or discrete-time) normalizing flows with classical sampling methods, such as MCMC, AIS, and SMC (Wu et al., 2020; Arbel et al., 2021; Matthews et al., 2022). Most of these methods rely on the reverse KL divergence that suffers from mode collapse. To combat this issue, the underlying continuity equation (and Hamilton-Jacobi-Bellman equations in case of optimal transport) have been leveraged for the learning problem (Ruthotto et al., 2020; Máté & Fleuret, 2023; Sun et al., 2024). However, in all the above cases, one needs to either restrict model expressivity or rely on costly computations of divergences (in continuous time) or Jacobian determinants (in discrete time). Our Prop. 3.4 shows that, in the stochastic case, the discrepancy in the corresponding Fokker-Planck equation – an expression involving divergences and Laplacians – can be approximated by detailed balance divergences, which require no differentiation.

**Diffusion-based samplers.** Motivated by (annealed) Langevin dynamics and diffusion models, there is growing interest in the development of SDEs controlled by neural networks, also known as neural SDEs, for sampling. This covers methods based on *Schrödinger (Half-)bridges* (Zhang & Chen, 2022), diffusion models (Vargas et al., 2023; Berner et al., 2022), and annealed flows (Vargas et al., 2024). These methods can be interpreted as special cases of stochastic bridges, aiming at finding a time-reversal between two SDEs starting at the prior and target distributions (Vargas et al., 2024; Richter & Berner, 2024). In particular, this allows to consider general divergences between the associated measures on the SDE trajectories, such as the log-variance divergence (Richter et al., 2020; Nüsken & Richter, 2021). We note that there has also been some work on combining classical sampling methods with diffusion models (Phillips et al., 2024; Doucet et al., 2022).

**GFlowNets.** GFlowNets are originally defined in discrete space (Bengio et al., 2023), but were generalized to general measure spaces in (Lahlou et al., 2023), who proved the correctness of objectives in continuous time and experimented with using them to train diffusion models as samplers. However, the connection between GFlowNets and diffusion models had already been made informally by Malkin et al. (2023) for samplers of Boltzmann distributions and by Zhang et al. (2023) for maximum-likelihood training, and the latter showed a connection between detailed balance and sliced score matching, which has a similar flavor to our Prop. 3.4. GFlowNets are, in principle, more general than diffusion models with Gaussian noising, as the state space may change between time steps and the transition density does not need to be Gaussian, which has been taken advantage of in some applications (Volokhova et al., 2024; Phillips & Cipcigan, 2024).

**Accelerated integrators for diffusion models.** We remark that there has been great interest in developing accelerated sampling methods for diffusion models and the related continuous normalizing flows (*e.g.*, Shaul et al., 2024; Pandey et al., 2024). In particular, one can consider higher-order integrators for the associated *probability flow ODE* (Song et al., 2021b) or integrate parts of the SDE analytically (Zhang & Chen, 2023). However, we note that this research is concerned with accelerating *inference*, not training, of diffusion models and thus orthogonal to our research. While Denker et al. (2024) and Venkatraman et al. (2024) explored subsampling strategies for the gradient estimators to reduce memory costs, they did not analyze the effect of randomized time steps. For generative modeling, one has access to samples from the target distribution, allowing the use of simulation-free denoising score matching for training. For sampling problems

without access to samples, diffusion-based methods, such as those outlined in the previous paragraphs, need to rely on costly simulation-based objectives. However, our findings show that we can significantly accelerate these simulations during training with a negligible drop in inference-time performance.

## B Theory details

### B.1 Assumptions

Throughout the paper, we assume that all SDEs admit densities of their time marginals (w.r.t. the Lebesgue measure) that are sufficiently smooth such that we have strong solutions to the corresponding Fokker-Planck equations (see, *e.g.*, Friedman (1964; 1975); Durrett (1984) for corresponding conditions). In particular, we assume that[4] $p_{\text{prior}}, p_{\text{target}} \in C^\infty(\mathbb{R}^d, \mathbb{R}_{>0})$ are bounded. Furthermore, we assume that $\mu \in C^\infty([0,1] \times \mathbb{R}^d, \mathbb{R}^d)$ for all drifts $\mu$, *i.e.*, they are infinitely differentiable, and satisfy a uniform (in time) Lipschitz condition, *i.e.*, there exists a constant $C$ such that for all $x, y \in \mathbb{R}^d$ and $t \in [0,1]$ it holds that

$$\|\mu(x,t) - \mu(y,t)\| \le C\|x - y\|. \tag{20}$$

Moreover, we assume that the diffusion rate satisfies that $\sigma \in C^\infty([0,1], \mathbb{R}_{>0})$. These conditions guarantee the existence of unique strong solutions to the considered SDEs (see, *e.g.*, Le Gall (2016)). They are also sufficient for all considered path measures to be equivalent and for Girsanov's theorem and Nelson's relation to hold. Moreover, they allow the definition of the forward and backward Itô integrals via limits of time discretizations that are independent of the specific sequence of refinements (Vargas et al., 2024).

While we use these assumptions to simplify the presentation, we note that they can be significantly relaxed. However, the above conditions hold for neural networks with MLP architectures whose activations are Lipschitz and infinitely differentiable, such as the GELU-activated models we consider in this paper.

### B.2 Formal definition of the MDP

We elaborate the definition of the MDP in §2.1.

- The state space is

$$\mathcal{S} = \{\bullet\} \cup \bigcup_{n=0}^{N} \underbrace{\{(x, t_n) : x \in \mathbb{R}^d\}}_{:=\mathcal{S}_n} \cup \{\bot\}, \tag{21}$$

  where $\bullet$ and $\bot$ are abstract initial and terminal states.
- The action space is $\mathcal{A} = \mathbb{R}^d$.
- The transition function $T \colon \mathcal{S} \times \mathcal{A} \to \mathcal{S}$ describing the deterministic effect of actions is given by

$$T(\bullet, a) = (a, t_0), \quad T((x, t_n), a) = \begin{cases} (a, t_{n+1}) & n < N \\ \bot & n = N \end{cases}, \quad T(\bot, a) = \bot. \tag{22}$$

- The reward is nonzero only for transitions from states in $\mathcal{S}_N$ to $\bot$ and is given by $R(x, t_N) = -\mathcal{E}(x)$.

It is arguably more natural from a control theory perspective to treat the addition of (*e.g.*, Gaussian) noise as stochasticity of the environment, making the policy deterministic. However, we choose to formulate integration as a constrained stochastic policy in a deterministic environment to allow flexibility in the form of the conditional distribution. We also note that the policy at $\bot$ is irrelevant since $\bot$ is an absorbing state.

### B.3 Numerical analysis

**Definition B.1** (Strong convergence)**.** A numerical scheme $\widehat{X} = (\widehat{X}_n)_{n=0}^{N}$ is called *strongly convergent* of order $\gamma$ if

$$\max_{n=0,\ldots,N} \mathbb{E}\left[\|\widehat{X}_n - X_{t_n}\|\right] \le C \left(\max_{n=0}^{N-1} \Delta t_n\right)^\gamma, \tag{23}$$

---

[4]Note that we also consider samplers using a Dirac delta prior, which can be treated by relaxing our conditions (Dai Pra, 1991). Under the policy given by (16), we can equivalently consider a (discrete-time) setting on the time interval $[t_1, 1]$ using a Gaussian prior with learned mean and variance $\sigma^2(t_0)\Delta t_0$.

where $0 < C < \infty$ is independent of $N \in \mathbb{N}$ and the time discretization $0 = t_0 < t_1 < \cdots < t_N = 1$.

**Definition B.2** (Weak convergence). *A numerical scheme $\widehat{X} = (\widehat{X}_n)_{n=0}^N$ is called* weakly *convergent of order $\gamma$ if*

$$\max_{n=0,\dots,N} \left\| \mathbb{E}[f(\widehat{X}_n)] - \mathbb{E}[f(X_{t_n})] \right\| \leq C \left( \max_{n=0}^{N-1} \Delta t_n \right)^{\gamma} \tag{24}$$

*for all functions $f$ in a suitable test class, where we consider $f \in C^{\infty}(\mathbb{R}^d, \mathbb{R})$ with at most polynomially growing derivatives. The constant $0 < C < \infty$ is independent of $N \in \mathbb{N}$ and the time discretization $0 = t_0 < t_1 < \cdots < t_N = 1$, but may depend on the class of test functions considered.*

Note that if $f$ is globally Lipschitz, then strong convergence implies weak convergence. The converse does not hold.

Let us also consider a continuous version $\iota(\widehat{X})$ of the numerical scheme $\widehat{X} = (\widehat{X}_n)_{n=0}^N$ defined by $\iota(\widehat{X})_{t_n} = \widehat{X}_n$ and linearly interpolating between the $t_n$, where we note that $\iota$ implicitly depends on the discretization. We can then define the pushforward $\iota_*\widehat{\mathbb{P}}$ of the distribution $\widehat{\mathbb{P}}$ of $\widehat{X}$ on the space of continuous functions $C([0,1], \mathbb{R}^d)$. We say that $\iota_*\widehat{\mathbb{P}}$ *converges weakly* to the path measure $\mathbb{P}$ of $X$ if for any bounded, continuous functional $f \colon C([0,1], \mathbb{R}^d) \to \mathbb{R}$ it holds that

$$\mathbb{E}_{X \sim \iota_*\widehat{\mathbb{P}}}[f(X)] \longrightarrow \mathbb{E}_{X \sim \mathbb{P}}[f(X)] \tag{25}$$

as $\max_n \Delta t_n \to 0$.

## B.4 Subtrajectory balance

Generalizing trajectory balance (7) and detailed balance (8), we can define divergences for subtrajectories of any length $k$ by multiplying the log-ratios appearing in (8) for several consecutive values of $n$, which through telescoping cancellation yields a *subtrajectory balance* divergence, defined for any $0 \leq n < n + k \leq N$ by

$$D_{\text{SubTB},n,n+k}^{\widehat{\mathbb{W}}}(\widehat{\mathbb{P}}, \widehat{\mathbb{Q}}, \widehat{p}) = \mathbb{E}_{\widehat{X} \sim \widehat{\mathbb{W}}} \left[ \log \left( \frac{\widehat{p}_n(\widehat{X}_n) \prod_{i=0}^{k-1} \overrightarrow{\pi}(\widehat{X}_{n+i+1} \mid \widehat{X}_{n+i})}{\widehat{p}_{n+k}(\widehat{X}_{n+k}) \prod_{i=0}^{k-1} \overleftarrow{\pi}(\widehat{X}_{n+i} \mid \widehat{X}_{n+i+1})} \right)^2 \right]. \tag{26}$$

The subtrajectory balance (SubTB) divergence generalizes detailed balance and trajectory balance, as one has

$$D_{\text{SubTB},n,n+1}^{\widehat{\mathbb{W}}}(\widehat{\mathbb{P}}, \widehat{\mathbb{Q}}, \widehat{p}) = D_{\text{DB}}^{n,\widehat{\mathbb{W}}}(\widehat{\mathbb{P}}, \widehat{\mathbb{Q}}, \widehat{p}) \quad \text{and} \quad D_{\text{SubTB},0,N}^{\widehat{\mathbb{W}}}(\widehat{\mathbb{P}}, \widehat{\mathbb{Q}}, \widehat{p}) = D_{\text{TB}}^{\widehat{\mathbb{W}}}(\widehat{\mathbb{P}}, \widehat{\mathbb{Q}}).$$

The SubTB divergence was introduced for GFlowNets by Malkin et al. (2022) and studied as a learning scheme, in which the divergences with different values of $k$ are appropriately weighted, by Madan et al. (2023). SubTB was tested in the diffusion sampling case by Zhang et al. (2024), although Sendera et al. (2024) found that it is, in general, not more effective than TB while being substantially more computationally expensive.

## B.5 Inductive bias on density estimates

We describe the inductive bias on density estimates used in the FL-DB learning objective. While normally one parametrizes the log-density as a neural network taking $x$ and $t$ as input:

$$\log \widehat{p}(x, t) = \text{NN}_\theta(x, t),$$

the inductive bias proposed by Wu et al. (2020); Máté & Fleuret (2023) and studied earlier for GFlowNet diffusion samplers by Zhang et al. (2024); Sendera et al. (2024) writes

$$\log \widehat{p}(x, t) = -t\mathcal{E}(x) + (1 - t) \log p_{\text{ref}}(x) + \text{NN}_\theta(x, t),$$

where $p_{\text{ref}}(\cdot, t)$ is the marginal density at time $t$ of the uncontrolled process, *i.e.*, the SDE (1) that sets $\overrightarrow{\mu} \equiv 0$ and has initial condition $p_{\text{prior}}$. Thus a correction is learned to an estimated log-density that interpolates between the prior at $t = 0$ and the target at $t = 1$.

The acronym 'FL-' stands for 'forward-looking', referring to the technique studied for GFlowNets by Pan et al. (2023) and understood as a form of reward-shaping scheme in Deleu et al. (2024).

## B.6   Proofs of results from the main text

**Proposition B.3** (Convergence of functionals). *If $\mathbb{P}, \mathbb{Q}, \mathbb{W}$ are path measures of three forward-time SDEs, the drifts of the SDEs and their derivatives are bounded, and $f \in C^\infty(\mathbb{R}, \mathbb{R})$ has at most polynomially growing derivatives, then*

$$\mathbb{E}_{\widehat{X} \sim \widehat{\mathbb{W}}} \left[ f \left( \log \frac{\mathrm{d}\widehat{\mathbb{P}}}{\mathrm{d}\widehat{\mathbb{Q}}}(\widehat{X}) \right) \right] \xrightarrow{\max_n \Delta t_n \to 0} \mathbb{E}_{X \sim \mathbb{W}} \left[ f \left( \log \frac{\mathrm{d}\mathbb{P}}{\mathrm{d}\mathbb{Q}}(X) \right) \right].$$

*Proof of Prop. 3.2.* As shown in the proof of Lemma B.7, $\log \frac{\mathrm{d}\widehat{\mathbb{P}}}{\mathrm{d}\widehat{\mathbb{Q}}}(\widehat{X})$ can be viewed as a component of the Euler-Maruyama integration of an Itô process (with space-dependent diffusion) evaluated at time 1. Given our assumptions on the coefficient functions (see also §B.1), the result follows by weak convergence; see, e.g., Kloeden & Platen (1992). □

**Proposition B.4** (Asymptotic consistency of TB and VarGrad). *Under the assumptions of Prop. 3.2, the divergences $D_{\mathrm{TB}}^{\widehat{\mathbb{W}}}(\widehat{\mathbb{P}}, \widehat{\mathbb{Q}})$ and $D_{\mathrm{LV}}^{\widehat{\mathbb{W}}}(\widehat{\mathbb{P}}, \widehat{\mathbb{Q}})$ converge to $D_{\mathrm{TB}}^{\mathbb{W}}(\mathbb{P}, \mathbb{Q})$ and $D_{\mathrm{LV}}^{\mathbb{W}}(\mathbb{P}, \mathbb{Q})$, respectively.*

*Proof of Prop. 3.3.* Immediate from Prop. 3.2, taking $f(x) = x^2$ and $f(x) = x$. □

**Proposition B.5** (Asymptotic equality of DB and FPE). *Under the smoothness conditions in Lemma B.8, $\overrightarrow{\mu}, \overleftarrow{\mu}, \widehat{p}$ jointly satisfy Nelson's identity ($\overleftarrow{\mu} = \overrightarrow{\mu} - \sigma^2 \nabla \log \widehat{p}$) at $(x_t, t)$ if and only if*

$$\lim_{h \to 0} \left[ \frac{1}{\sqrt{h}} \Delta_{t \to t+h}(x_t, x_{t+h}) \right] = 0 \quad \text{for almost every } z,$$

*where $x_{t+h} := x_t + \overrightarrow{\mu}(x_t, t)h + \sigma(t)\sqrt{h}z$. If in addition*

$$\lim_{h \to 0} \mathbb{E}_{z \sim \mathcal{N}(0, I_d)} \left[ \frac{1}{h} \Delta_{t \to t+h}(x_t, x_{t+h}) \right] = 0,$$

*then the Fokker-Plank equation is satisfied at $(x_t, t)$. If both conditions hold at all $(x_t, t) \in \mathbb{R}^d \times (0, 1)$, then $\overrightarrow{\mu}, \overleftarrow{\mu}$ define a pair of time-reversed processes with marginal density $\widehat{p}$.*

*Proof of Prop. 3.4.* We write $\widehat{p}_t(x), \overrightarrow{\mu}_t(x), \sigma_t$ for $\widehat{p}(x, t), \overrightarrow{\mu}(x, t), \sigma(t)$ for convenience. By Lemma B.8, the first condition implies that for almost all $z$,

$$\langle z, \sigma_t^2 \nabla \log \widehat{p}_t(x_t) - (\overrightarrow{\mu}_t(x_t) - \overleftarrow{\mu}_t(x_t)) \rangle = 0, \tag{27}$$

which implies Nelson's identity at $(x_t, t)$, while the second condition implies that

$$\partial_t \log \widehat{p}_t(x_t) + \langle \overrightarrow{\mu}_t(x_t), \nabla \log \widehat{p}_t(x_t) \rangle + \langle \nabla, \overleftarrow{\mu}_t(x_t) \rangle + \frac{\sigma_t^2}{2} \left( \Delta \log \widehat{p}_t(x_t) - \left\| \frac{\overrightarrow{\mu}_t(x_t) - \overleftarrow{\mu}_t(x_t)}{\sigma_t^2} \right\|^2 \right) = 0. \tag{28}$$

Substituting the expression (27) into (28) and simplifying, we get

$$\partial_t \log \widehat{p}_t(x_t) = -\langle \nabla, \overrightarrow{\mu}_t(x_t) \rangle - \langle \overrightarrow{\mu}_t(x_t), \nabla \log \widehat{p}_t(x_t) \rangle + \frac{\sigma_t^2}{2} \left( \Delta \log \widehat{p}_t(x_t) + \| \nabla \log \widehat{p}_t(x_t) \|^2 \right),$$

which gives exactly the logarithmic form of the Fokker-Planck equation. □

**Proposition B.6** (DB and FPE for Brownian bridges). *The results of Prop. 3.4 hold if $\sigma(t)$ is constant and $\overleftarrow{\pi}$ is the discrete-time reversal of the Euler-Maruyama discretization of the process $\mathrm{d}X_t = \sigma(t)\,\mathrm{d}W_t$, $p_{\mathrm{prior}}(x) = \mathcal{N}(x; 0, \sigma_0 I_d)$.*

*Proof of Prop. 3.5.* Using the changes of variables $x \mapsto \sigma x$ followed $t \mapsto t - \sigma_0$, it suffices to show this for $\sigma_0 = 0, \sigma = 1$, making the process a standard Brownian motion (the change of bounds for $t$ is insubstantial as the conditions are local in time).

Let $\overleftarrow{\pi}$ be the backward policy as originally defined. The reverse drift is $\overleftarrow{\mu}(x,t) = \frac{x}{t}$, so we have

$$\overleftarrow{\pi}(x_t \mid x_{t+h}) = \mathcal{N}\left(x_t; \frac{t}{t+h}x_{t+h}, h\right).$$

Let $\overleftarrow{\pi}'$ be the discrete-time reversal of the forward-discretized Brownian motion. By elementary properties of Gaussians, we have

$$\overleftarrow{\pi}'(x_t \mid x_{t+h}) = \mathcal{N}\left(x_t; \frac{t}{t+h}x_{t+h}, \frac{t}{t+h}h\right).$$

Let $\Delta_{t \to t+h}(x_t, x_{t+h})$ and $\Delta'_{t \to t+h}(x_t, x_{t+h})$ be the discrepancies (18)) defined using $\overleftarrow{\pi}$ and $\overleftarrow{\pi}'$, respectively. We will show that replacing $\Delta$ by $\Delta'$ does not affect the asymptotics in Prop. 3.4.

We have

$$\Delta_{t \to t+h}(x_t, x_{t+h}) - \Delta'_{t \to t+h}(x_t, x_{t+h}) = \log \overleftarrow{\pi}'(x_t \mid x_{t+h}) - \log \overleftarrow{\pi}(x_t \mid x_{t+h})$$

$$= \frac{-1}{2}\left[d \log \frac{t}{t+h} + \left\|x_t - \frac{t}{t+h}x_{t+h}\right\|^2 \left(\frac{1}{\frac{t}{t+h}h} - \frac{1}{h}\right)\right]$$

$$= \frac{-1}{2}\left[d \log \left(1 - \frac{h}{t+h}\right) + \frac{1}{t}\left\|x_t - x_{t+h} + \frac{h}{t+h}x_{t+h}\right\|^2\right].$$

Setting $x_{t+h} = x_t + \overrightarrow{\mu}_t(x_t)h + \sqrt{h}z$, the above becomes

$$\frac{-1}{2}\left[-\frac{h}{t}d + O(h^2) + \frac{1}{t}\left(h\|z\|^2 + O(h^{3/2})\right)\right].$$

For fixed $z$, the $\sqrt{h}$-order asymptotics of this expression vanish. In expectation over $z \sim \mathcal{N}(0, I_d)$, the $h$-order asymptotics vanish because $\mathbb{E}_{z \sim \mathcal{N}(0, I_d)}\left[\|z\|^2\right] = d$. $\qquad \square$

### B.7 Technical lemmas

**Lemma B.7** (Convergence of Radon-Nikodym derivatives). *(a) Let $\mathbb{P}^{(1)}$ and $\mathbb{P}^{(2)}$ be the path space measures defined by SDEs of the form (9) with initial conditions $p_{\text{prior}}^{(1),(2)}$ and drifts $\overrightarrow{\mu}^{(1),(2)}$. Let $\widehat{\mathbb{P}}^{(1),(2)}$ be the Euler-Maruyama-discretized measures with respect to a time discretization $(t_n)_{n=0}^N$. For $\mathbb{P}^{(2)}$-almost every $X \in C([0,1], \mathbb{R}^d)$, $\frac{d\widehat{\mathbb{P}}^{(1)}}{d\widehat{\mathbb{P}}^{(2)}}(X_{t_0,\dots,t_N}) \to \frac{d\mathbb{P}^{(1)}}{d\mathbb{P}^{(2)}}(X)$ as $\max_n \Delta t_n \to 0$, where $X_{t_0,\dots,t_N}$ is the restriction of $X$ to the times $t_0, \dots, t_N$.*

*(b) The same is true for a path space measure $\mathbb{P}$ defined by a forward SDE with initial conditions and a measure $\mathbb{Q}$ defined by a reverse SDE with terminal conditions: if $\widehat{\mathbb{P}}$ and $\widehat{\mathbb{Q}}$ are the discrete-time processes given by Euler-Maruyama and reverse Euler-Maruyama integration, respectively, then for $\mathbb{Q}$-almost every $X \in C([0,1], \mathbb{R}^d)$, as $\max_n \Delta t_n \to 0$, $\frac{d\widehat{\mathbb{P}}}{d\widehat{\mathbb{Q}}}(X_{t_0,\dots,t_N}) \to \frac{d\mathbb{P}}{d\mathbb{Q}}(X)$.*

*Proof.* We first show (a). We have

$$
\begin{aligned}
\log \frac{\mathrm{d}\widehat{\mathbb{P}}^{(1)}}{\mathrm{d}\widehat{\mathbb{P}}^{(2)}}(X_{t_0,\ldots,t_N}) &= \log \frac{p_{\mathrm{prior}}^{(1)}(X_0) \prod_{n=0}^{N-1} \overrightarrow{\pi}_n(X_{t_{n+1}} \mid X_{t_n})}{p_{\mathrm{prior}}^{(2)}(X_0) \prod_{n=0}^{N-1} \overrightarrow{\pi}_n(X_{t_{n+1}} \mid X_{t_n})} \\
&= \log \frac{p_{\mathrm{prior}}^{(1)}(X_0)}{p_{\mathrm{prior}}^{(2)}(X_0)} + \sum_{n=0}^{N-1} \log \frac{\mathcal{N}(X_{t_{n+1}}; X_{t_n} + \overrightarrow{\mu}^{(1)}(X_{t_n}, t_n)\Delta t_n, \sigma(t_n)^2 \Delta t_n)}{\mathcal{N}(X_{t_{n+1}}; X_{t_n} + \overrightarrow{\mu}^{(2)}(X_{t_n}, t_n)\Delta t_n, \sigma(t_n)^2 \Delta t_n)} \\
&= \log \frac{p_{\mathrm{prior}}^{(1)}(X_0)}{p_{\mathrm{prior}}^{(2)}(X_0)} + \sum_{n=0}^{N-1} \bigg[ -\frac{\|\overrightarrow{\mu}^{(1)}(X_{t_n}, t_n)\|^2 - \|\overrightarrow{\mu}^{(2)}(X_{t_n}, t_n)\|^2}{2\sigma(t_n)^2} \Delta t_n \\
&\qquad\qquad + \frac{\overrightarrow{\mu}^{(1)}(X_{t_n}, t_n) - \overrightarrow{\mu}^{(2)}(X_{t_n}, t_n)}{\sigma(t_n)^2} \cdot (X_{t_{n+1}} - X_{t_n}) \bigg].
\end{aligned}
\tag{29}
$$

This is precisely the (Riemann) sum for the integral defining the continuous-time Radon-Nikodym derivative (13); by continuity and our assumptions in §B.1, the sum approaches the integral as $\max_n \Delta t_n \to 0$.

We now show (b) assuming (a). Let $\mathbb{P}^0$ be the path measure defined by Gaussian $\mathcal{N}(0, I)$ initial conditions and drift 0 and $\widehat{\mathbb{P}}^0$ its discretization. Similarly, let $\mathbb{Q}^0$ be defined by Gaussian terminal conditions and zero reverse drift and let $\widehat{\mathbb{Q}}^0$ be its reverse-time discretization. By absolute continuity, we have

$$
\frac{\mathrm{d}\mathbb{P}}{\mathrm{d}\mathbb{Q}}(X) = \frac{\mathrm{d}\mathbb{P}/\mathrm{d}\mathbb{P}^0(X)}{\mathrm{d}\mathbb{Q}/\mathrm{d}\mathbb{Q}^0(X)} \frac{\mathrm{d}\mathbb{P}^0}{\mathrm{d}\mathbb{Q}^0}(X), \qquad \frac{\mathrm{d}\widehat{\mathbb{P}}}{\mathrm{d}\widehat{\mathbb{Q}}}(X_{t_0,\ldots,t_N}) = \frac{\mathrm{d}\widehat{\mathbb{P}}/\mathrm{d}\widehat{\mathbb{P}}^0(X_{t_0,\ldots,t_N})}{\mathrm{d}\widehat{\mathbb{Q}}/\mathrm{d}\widehat{\mathbb{Q}}^0(X_{t_0,\ldots,t_N})} \frac{\mathrm{d}\widehat{\mathbb{P}}^0}{\mathrm{d}\widehat{\mathbb{Q}}^0}(X_{t_0,\ldots,t_N}).
$$

By (a), $\mathrm{d}\widehat{\mathbb{P}}/\mathrm{d}\widehat{\mathbb{P}}^0(X_{t_0,\ldots,t_N}) \to \mathrm{d}\mathbb{P}/\mathrm{d}\mathbb{P}^0(X)$, and similarly for $\mathbb{Q}$. It remains to show that $\log \mathrm{d}\widehat{\mathbb{P}}^0/\mathrm{d}\widehat{\mathbb{Q}}^0(X_{t_0,\ldots,t_N}) \to \log \mathrm{d}\mathbb{P}^0/\mathrm{d}\mathbb{Q}^0(X) = \log \mathcal{N}(X_0; 0, I) - \log \mathcal{N}(X_1; 0, I)$. Indeed, we have

$$
\begin{aligned}
\log \mathrm{d}\widehat{\mathbb{P}}^0/\mathrm{d}\widehat{\mathbb{Q}}^0(X_{t_0,\ldots,t_N}) &= \log \frac{\mathcal{N}(X_0; 0, I)}{\mathcal{N}(X_1; 0, I)} + \sum_{n=1}^{N} \log \frac{\mathcal{N}(X_{t_n}; X_{t_{n-1}}, \sigma(t_{n-1})\Delta t_{n-1})}{\mathcal{N}(X_{t_{n-1}}; X_{t_n}, \sigma(t_n)\Delta t_{n-1})} \\
&= \log \frac{\mathcal{N}(X_0; 0, I)}{\mathcal{N}(X_1; 0, I)} + \sum_{n=1}^{N} \bigg[ \frac{\|X_{t_n} - X_{t_{n-1}}\|^2}{2\Delta t_{n-1}} \bigg( \frac{1}{\sigma(t_n)^2} - \frac{1}{\sigma(t_{n-1})^2} \bigg) \\
&\qquad\qquad\qquad\qquad\qquad + d \log \frac{\sigma(t_n)}{\sigma(t_{n-1})} \bigg] \\
&\xrightarrow{\text{a.s.}} \log \frac{\mathcal{N}(X_0; 0, I)}{\mathcal{N}(X_1; 0, I)} + d \log \frac{\sigma(1)}{\sigma(0)} + \underbrace{\int_0^1 \frac{\mathrm{d}\sigma(t)^2}{2} \mathrm{d}\sigma(t)^{-2}}_{=-\mathrm{d}(d \log \sigma(t))} \\
&= \log \frac{\mathcal{N}(X_0; 0, I)}{\mathcal{N}(X_1; 0, I)}.
\end{aligned}
\tag{30}
$$

which coincides with the continuous-time Radon-Nikodym derivative. $\qquad\square$

**Lemma B.8** (Continuous-time asymptotics of the DB discrepancy)**.** *Let us define the abbreviations $\widehat{p}_t(x)$, $\overrightarrow{\mu}_t(x)$, $\sigma_t$ to refer to $\widehat{p}(x, t)$, $\overrightarrow{\mu}(x, t)$, $\sigma(t)$. Suppose that $\overleftarrow{\mu}_t$ and $\overleftarrow{\mu}_t$ are continuously differentiable in $x$ and once in $t$ and that $\log \widehat{p}_t$ is continuously differentiable once in $t$ and twice in $x$.*

*(a) For a given $z$, the asymptotics of the DB discrepancy at $(x_t, t)$ are of order $\sqrt{h}$ and are given by*

$$
\lim_{h \to 0} \bigg[ \frac{1}{\sqrt{h}} \Delta_{t \to t+h}(x_t, x_t + \overrightarrow{\mu}_t(x_t)h + \sigma_t z) \bigg] = \sigma_t^{-1} \langle z, \sigma_t^2 \nabla \log \widehat{p}_t(x_t) - (\overrightarrow{\mu}_t(x_t) - \overleftarrow{\mu}_t(x_t)) \rangle.
$$

*(b) The expectation of the DB discrepancy over the forward policy (*i.e.*, over $z \sim \mathcal{N}(0, I)$) is asymptotically of order $h$, with leading term*

$$\lim_{h \to 0} \mathbb{E}_{x_{t+h} \sim \overrightarrow{\pi}(x_{t+h}|x_t)} \left[ \frac{1}{h} \Delta_{t \to t+h}(x_t, x_{t+h}) \right]$$

$$= \partial_t \log \widehat{p}_t(x_t) + \langle \overrightarrow{\mu}_t(x_t), \nabla \log \widehat{p}_t(x_t) \rangle + \langle \nabla, \overleftarrow{\mu}_t(x_t) \rangle$$

$$+ \frac{\sigma_t^2}{2} \left( \Delta \log \widehat{p}_t(x_t) - \left\| \frac{\overrightarrow{\mu}_t(x_t) - \overleftarrow{\mu}_t(x_t)}{\sigma_t^2} \right\|^2 \right).$$

*Similarly, the expectation over the backward policy is*

$$\lim_{h \to 0} \mathbb{E}_{x_{t-h} \sim \overleftarrow{\pi}(x_{t-h}|x_t)} \left[ \frac{1}{h} \Delta_{t-h \to t}(x_{t-h}, x_t) \right]$$

$$= \partial_t \log \widehat{p}_t(x_t) + \langle \overleftarrow{\mu}_t(x_t), \nabla \log \widehat{p}_t(x_t) \rangle + \langle \nabla, \overrightarrow{\mu}_t(x_t) \rangle$$

$$- \frac{\sigma_t^2}{2} \left( \Delta \log \widehat{p}_t(x_t) - \left\| \frac{\overrightarrow{\mu}_t(x_t) - \overleftarrow{\mu}_t(x_t)}{\sigma_t^2} \right\|^2 \right).$$

*Proof.* We will simultaneously show (a) and the first part of (b). The second part of (b) is symmetric, by reversing time.

Identifying $x_{t+h}$ with $x_t + \overrightarrow{\mu}_t(x_t)h + \sigma_t \sqrt{h} z$, we will analyze the leading asymptotics of the DB discrepancy, i.e., $\Delta_{t \to t+h}(x_t, x_{t+h}) = \sqrt{h}\langle z, \dots \rangle + h(\dots) + \mathcal{O}(h^{3/2})$. The coefficient of $\sqrt{h}$ will be the scalar product of $z$ with a term that is independent of $z$ and equals the expression on the right side in (a), and thus vanishes in expectation over $z$. The coefficient of $h$, in expectation over $z$, will equal the expression on the right side in (b).

We can show using Taylor expansions that

$$\log \frac{\widehat{p}_{t+h}(x_{t+h})}{\widehat{p}_t(x_t)} = \sqrt{h} \langle z, \sigma_t \nabla \log \widehat{p}_t(x_t) \rangle$$

$$+ h \left[ \partial_t \log \widehat{p}_t(x_t) + \langle \overrightarrow{\mu}_t(x_t), \nabla \log \widehat{p}_t(x_t) \rangle + \frac{1}{2} \sigma_t^2 \langle z, \nabla^2 \log \widehat{p}_t(x_t) z \rangle \right] + \mathcal{O}(h^{3/2}). \qquad (31)$$

Now we are going to analyze the second part of (18), which involves the policies. We have

$$\log \frac{\overleftarrow{\pi}(x_t \mid x_{t+h})}{\overrightarrow{\pi}(x_{t+h} \mid x_t)}$$

$$= \frac{-1}{2} \left[ \frac{\|x_t - x_{t+h} + \overleftarrow{\mu}_{t+h}(x_{t+h})h\|^2}{\sigma_{t+h}^2 h} - \frac{\|x_{t+h} - x_t - \overrightarrow{\mu}_t(x_t)h\|^2}{\sigma_t^2 h} + d \log \frac{2\pi \sigma_{t+h}^2}{2\pi \sigma_t^2} \right]$$

$$= \frac{-1}{2} \left[ \frac{\|x_t - x_{t+h} + \overleftarrow{\mu}_{t+h}(x_{t+h})h\|^2}{\sigma_{t+h}^2 h} \right] + \frac{\|\sigma_t \sqrt{h} z\|^2}{2\sigma_t^2 h} - d \log \frac{\sigma_{t+h}}{\sigma_t}.$$

$$= \frac{-1}{2} \left[ \frac{\|x_t - x_{t+h} + \overleftarrow{\mu}_{t+h}(x_{t+h})h\|^2}{\sigma_{t+h}^2 h} \right] + \frac{\|z\|^2}{2} - d h \partial_t(\log \sigma_t) + \mathcal{O}(h^2).$$

Next we will write

$$x_t - x_{t+h} + \overleftarrow{\mu}_{t+h}(x_{t+h})h = x_t - x_{t+h} + \overrightarrow{\mu}_t(x_t)h - (\overrightarrow{\mu}_t(x_t) - \overleftarrow{\mu}_{t+h}(x_{t+h}))h$$

$$= -\sigma_t \sqrt{h} z - (\overrightarrow{\mu}_t(x_t) - \overleftarrow{\mu}_{t+h}(x_{t+h}))h$$

and substitute this into the first term above, yielding

$$
\frac{-1}{2}\left[\frac{\|x_t - x_{t+h} + \overleftarrow{\mu}_{t+h}(x_{t+h})h\|^2}{\sigma_{t+h}^2 h}\right] + \frac{\|z\|^2}{2} - dh\partial_t(\log\sigma_t) + \mathcal{O}(h^2)
$$

$$
= \frac{-1}{2}\left[\frac{\|-\sigma_t\sqrt{h}z - (\overrightarrow{\mu}_t(x_t) - \overleftarrow{\mu}_{t+h}(x_{t+h}))h\|^2}{\sigma_{t+h}^2 h}\right] + \frac{\|z\|^2}{2} - dh\partial_t(\log\sigma_t) + \mathcal{O}(h^2)
$$

$$
= -\frac{\|\overrightarrow{\mu}_t(x_t) - \overleftarrow{\mu}_{t+h}(x_{t+h})\|^2}{2\sigma_{t+h}^2}h - \frac{\langle\sigma_t z, \overrightarrow{\mu}_t(x_t) - \overleftarrow{\mu}_{t+h}(x_{t+h})\rangle}{\sigma_{t+h}^2}\sqrt{h}
$$

$$
\quad - \frac{\sigma_t^2\|z\|^2}{2\sigma_{t+h}^2} + \frac{\|z\|^2}{2} - dh\partial_t(\log\sigma_t) + \mathcal{O}(h^2)
$$

$$
= \sqrt{h}\left[\langle z, -\sigma_t^{-1}(\overrightarrow{\mu}_t(x_t) - \overleftarrow{\mu}_t(x_t))\rangle + \frac{\sigma_t\langle z, \sigma_t\sqrt{h}\nabla\overleftarrow{\mu}_t(x_t)z\rangle}{\sigma_{t+h}^2}\right]
$$

$$
\quad + h\left[-\frac{\|\overrightarrow{\mu}_t(x_t) - \overleftarrow{\mu}_t(x_t)\|^2}{2\sigma_t^2} - d\partial_t(\log\sigma_t)\right] + \underbrace{\frac{\|z\|^2}{2}\left(1 - \frac{\sigma_t^2}{\sigma_{t+h}^2}\right)}_{=2\partial_t(\log\sigma_t)h + \mathcal{O}(h^2)} + \mathcal{O}(h^{3/2})
$$

$$
= \sqrt{h}\left\langle z, -\sigma_t^{-1}(\overrightarrow{\mu}_t(x_t) - \overleftarrow{\mu}_t(x_t))\right\rangle
$$

$$
\quad + h\left[-\frac{\|\overrightarrow{\mu}_t(x_t) - \overleftarrow{\mu}_t(x_t)\|^2}{2\sigma_t^2} + \langle z, \nabla\overleftarrow{\mu}_t(x_t)z\rangle - \left(\|z\|^2 - d\right)\partial_t(\log\sigma_t)\right] + \mathcal{O}(h^{3/2}).
$$

Combining with the terms in (31), we get that the coefficient of $\sqrt{h}$ is exactly as desired. For the coefficient of $h$, and the terms of the form $\langle z, \dots\rangle$ and $\|z\|^2 - d$ vanish in expectation over $z$. For the terms that are quadratic in $z$, Hutchinson's formula implies that

$$
\mathbb{E}_{z\sim\mathcal{N}(0,I)}\left[\langle z, \nabla\overleftarrow{\mu}_t(x_t)z\rangle\right] = \langle\nabla, \overleftarrow{\mu}_t(x_t)\rangle,
$$
$$
\mathbb{E}_{z\sim\mathcal{N}(0,I)}\left[\langle z, \nabla^2\log\widehat{p}_t(x_t)z\rangle\right] = \Delta\log\widehat{p}_t(x_t).
$$

Putting these identities together, we obtain that

$$
\lim_{h\to 0}\mathbb{E}_{x_{t+h}\sim\overrightarrow{\pi}(x_{t+h}|x_t)}\left[\frac{1}{h}\Delta_{t\to t+h}(x_t, x_{t+h})\right]
$$

$$
= \partial_t\log\widehat{p}_t(x_t) + \langle\overrightarrow{\mu}_t(x_t), \nabla\log\widehat{p}_t(x_t)\rangle + \frac{1}{2}\sigma_t^2\Delta\log\widehat{p}_t(x_t) - \frac{\|\overrightarrow{\mu}_t(x_t) - \overleftarrow{\mu}_t(x_t)\|^2}{2\sigma_t^2} + \langle\nabla, \overleftarrow{\mu}_t(x_t)\rangle,
$$

which is equivalent to the expression in (b). $\qquad\square$

## C Experiment details

### C.1 Training settings

All models are trained for 25,000 steps using settings identical to those suggested by Sendera et al. (2024) (https://github.com/GFNOrg/gfn-diffusion). For DB, we use the same learning rates as for SubTB ($10^{-3}$ for the drift and $10^{-2}$ for the flow function), and for PIS, $10^{-3}$ or $10^{-4}$ depending on its stability in the specific case.

Training times are measured by wall time of execution on a large shared cluster, primarily on RTX8000 GPUs. Although all runs were assigned by the same job scheduler, some variability in results is inevitable due to inconsistent hardware.

# D    Additional results

## D.1    Additional metrics and objectives

In Table 2, we show extended results on the four unconditional sampling benchmarks from Sendera et al. (2024), reporting the ELBO $\log \widehat{Z}$ and importance-weighted ELBO $\log \widehat{Z}^{\mathrm{RW}}$. Specifically, the two are computed as

$$
\begin{aligned}
\log \widehat{Z} &:= \frac{1}{K} \sum_{i=1}^{K} \left[ -\mathcal{E}(\widehat{X}_N^{(i)}) + \log \frac{\widehat{\mathbb{Q}}(\widehat{X}^{(i)} \mid \widehat{X}_N^{(i)})}{\widehat{\mathbb{P}}(\widehat{X}^{(i)})} \right] = \log Z + \frac{1}{K} \sum_{i=1}^{K} \left[ \log \frac{\widehat{\mathbb{Q}}(\widehat{X}^{(i)})}{\widehat{\mathbb{P}}(\widehat{X}^{(i)})} \right], \\
\log \widehat{Z}^{\mathrm{RW}} &:= \log \frac{1}{K} \sum_{i=1}^{K} \exp \left[ -\mathcal{E}(\widehat{X}_N^{(i)}) + \log \frac{\widehat{\mathbb{Q}}(\widehat{X}^{(i)} \mid \widehat{X}_N^{(i)})}{\widehat{\mathbb{P}}(\widehat{X}^{(i)})} \right] = \log Z + \log \frac{1}{K} \sum_{i=1}^{K} \left[ \frac{\widehat{\mathbb{Q}}(\widehat{X}^{(i)})}{\widehat{\mathbb{P}}(\widehat{X}^{(i)})} \right],
\end{aligned}
\tag{32}
$$

where $\widehat{X}^{(1)}, \ldots \widehat{X}^{(K)} \sim \widehat{\mathbb{P}}$ and we note that $\mathbb{E}[\log \widehat{Z}] = \log Z - D_{\mathrm{KL}}(\widehat{\mathbb{P}}, \widehat{\mathbb{Q}}) \leq \log Z$ and $\mathbb{E}[Z^{\mathrm{RW}}] = Z$. We take $K = 2000$ samples and report the difference between the ground truth $\log Z$ and the ELBO when $\log Z$ is known.

These results are consistent with the conclusions in the main text. Notably, when combined with local search, coarse nonuniform discretizations continue to show results comparable to those of 100-step training discretization in most cases. Table 3 shows results on two additional target energies and on the conditional VAE task.

Table 2: ELBOs and IS-ELBOs on **25GMM**, **Funnel**, and **Manywell** (absolute error from the true value).

### 25GMM ($d = 2$)

| Training discretization → | 10-step random | | | | 10-step equidistant | | | | 10-step uniform | | | | 100-step uniform | |
|---|---|---|---|---|---|---|---|---|---|---|---|---|---|---|
| Evaluation steps → | 10 | | 100 | | 10 | | 100 | | 10 | | 100 | | 100 | |
| Algorithm ↓ Metric → | $\Delta \log Z$ | $\Delta \log Z^{\mathrm{RW}}$ | $\Delta \log Z$ | $\Delta \log Z^{\mathrm{RW}}$ | $\Delta \log Z$ | $\Delta \log Z^{\mathrm{RW}}$ | $\Delta \log Z$ | $\Delta \log Z^{\mathrm{RW}}$ | $\Delta \log Z$ | $\Delta \log Z^{\mathrm{RW}}$ | $\Delta \log Z$ | $\Delta \log Z^{\mathrm{RW}}$ | $\Delta \log Z$ | $\Delta \log Z^{\mathrm{RW}}$ |
| PIS | $2.40_{\pm0.10}$ | $1.02_{\pm0.09}$ | $1.56_{\pm0.10}$ | $0.93_{\pm0.16}$ | $2.39_{\pm0.11}$ | $0.97_{\pm0.10}$ | $1.51_{\pm0.09}$ | $1.01_{\pm0.09}$ | $2.43_{\pm0.12}$ | $0.85_{\pm0.58}$ | $5.62_{\pm0.32}$ | $1.03_{\pm0.14}$ | $1.65_{\pm0.30}$ | $1.12_{\pm0.20}$ |
| TB | $2.10_{\pm0.05}$ | $1.02_{\pm0.05}$ | $1.23_{\pm0.03}$ | $1.03_{\pm0.03}$ | $2.10_{\pm0.04}$ | $0.96_{\pm0.14}$ | $1.22_{\pm0.03}$ | $1.04_{\pm0.03}$ | $2.10_{\pm0.03}$ | $0.99_{\pm0.11}$ | $8.77_{\pm0.69}$ | $1.02_{\pm0.96}$ | $1.13_{\pm0.01}$ | $1.02_{\pm0.01}$ |
| TB + LS | $1.71_{\pm0.06}$ | $0.02_{\pm0.17}$ | $0.47_{\pm0.06}$ | $\mathbf{0.002_{\pm0.04}}$ | $1.71_{\pm0.04}$ | $0.16_{\pm0.07}$ | $0.42_{\pm0.03}$ | $0.03_{\pm0.02}$ | $1.67_{\pm0.06}$ | $0.05_{\pm0.02}$ | $10.38_{\pm2.78}$ | $1.87_{\pm0.77}$ | $0.16_{\pm0.01}$ | $\mathbf{0.0004_{\pm0.01}}$ |
| VarGrad | $2.12_{\pm0.04}$ | $1.04_{\pm0.04}$ | $1.22_{\pm0.01}$ | $1.04_{\pm0.01}$ | $2.09_{\pm0.03}$ | $1.04_{\pm0.01}$ | $1.19_{\pm0.03}$ | $1.03_{\pm0.01}$ | $2.12_{\pm0.02}$ | $1.02_{\pm0.04}$ | $9.13_{\pm0.87}$ | $0.92_{\pm1.19}$ | $1.12_{\pm0.01}$ | $1.02_{\pm0.01}$ |
| VarGrad + LS | $1.68_{\pm0.07}$ | $0.04_{\pm0.09}$ | $0.37_{\pm0.06}$ | $0.02_{\pm0.02}$ | $1.67_{\pm0.01}$ | $0.07_{\pm0.07}$ | $\mathbf{0.33_{\pm0.07}}$ | $0.02_{\pm0.01}$ | $1.62_{\pm0.04}$ | $0.06_{\pm0.07}$ | $8.25_{\pm0.95}$ | $1.11_{\pm0.24}$ | $\mathbf{0.15_{\pm0.004}}$ | $0.01_{\pm0.01}$ |
| PIS + LP | $2.80_{\pm0.07}$ | $1.02_{\pm0.17}$ | $1.98_{\pm0.06}$ | $0.10_{\pm0.42}$ | $2.77_{\pm0.10}$ | $1.00_{\pm0.21}$ | $1.94_{\pm0.03}$ | $0.05_{\pm0.30}$ | $2.77_{\pm0.08}$ | $1.00_{\pm0.20}$ | $3.49_{\pm0.08}$ | $0.14_{\pm1.24}$ | $1.76_{\pm0.02}$ | $0.43_{\pm0.45}$ |
| TB + LP | $1.57_{\pm0.05}$ | $0.03_{\pm0.18}$ | $0.32_{\pm0.02}$ | $0.02_{\pm0.05}$ | $1.56_{\pm0.03}$ | $0.01_{\pm0.16}$ | $0.36_{\pm0.06}$ | $0.03_{\pm0.03}$ | $2.70_{\pm2.33}$ | $0.11_{\pm0.33}$ | $5.30_{\pm0.80}$ | $0.43_{\pm0.47}$ | $0.16_{\pm0.01}$ | $0.01_{\pm0.01}$ |
| TB + LS + LP | $1.78_{\pm0.10}$ | $0.02_{\pm0.08}$ | $0.41_{\pm0.06}$ | $0.02_{\pm0.04}$ | $1.82_{\pm0.01}$ | $0.08_{\pm0.06}$ | $0.43_{\pm0.05}$ | $0.07_{\pm0.08}$ | $1.68_{\pm0.09}$ | $0.05_{\pm0.02}$ | $8.37_{\pm1.50}$ | $1.50_{\pm0.46}$ | $0.16_{\pm0.01}$ | $0.01_{\pm0.01}$ |
| VarGrad + LP | $1.59_{\pm0.04}$ | $0.03_{\pm0.08}$ | $0.35_{\pm0.06}$ | $\mathbf{0.01_{\pm0.02}}$ | $1.46_{\pm0.005}$ | $0.07_{\pm0.06}$ | $0.32_{\pm0.04}$ | $0.04_{\pm0.01}$ | $1.53_{\pm0.01}$ | $0.01_{\pm0.01}$ | $5.52_{\pm0.80}$ | $0.53_{\pm0.54}$ | $0.15_{\pm0.01}$ | $\mathbf{0.003_{\pm0.01}}$ |
| VarGrad + LS + LP | $1.68_{\pm0.09}$ | $0.02_{\pm0.08}$ | $0.26_{\pm0.02}$ | $\mathbf{0.01_{\pm0.01}}$ | $1.69_{\pm0.05}$ | $0.07_{\pm0.06}$ | $\mathbf{0.24_{\pm0.01}}$ | $\mathbf{0.01_{\pm0.01}}$ | $1.64_{\pm0.06}$ | $0.04_{\pm0.07}$ | $7.07_{\pm1.50}$ | $0.90_{\pm0.85}$ | $0.16_{\pm0.01}$ | $0.01_{\pm0.005}$ |

### Funnel ($d = 10$)

| Training discretization → | 10-step random | | | | 10-step equidistant | | | | 10-step uniform | | | | 100-step uniform | |
|---|---|---|---|---|---|---|---|---|---|---|---|---|---|---|
| Evaluation steps → | 10 | | 100 | | 10 | | 100 | | 10 | | 100 | | 100 | |
| Algorithm ↓ Metric → | $\Delta \log Z$ | $\Delta \log Z^{\mathrm{RW}}$ | $\Delta \log Z$ | $\Delta \log Z^{\mathrm{RW}}$ | $\Delta \log Z$ | $\Delta \log Z^{\mathrm{RW}}$ | $\Delta \log Z$ | $\Delta \log Z^{\mathrm{RW}}$ | $\Delta \log Z$ | $\Delta \log Z^{\mathrm{RW}}$ | $\Delta \log Z$ | $\Delta \log Z^{\mathrm{RW}}$ | $\Delta \log Z$ | $\Delta \log Z^{\mathrm{RW}}$ |
| PIS | $1.11_{\pm0.01}$ | $0.59_{\pm0.03}$ | $\mathbf{0.72_{\pm0.02}}$ | $0.09_{\pm0.50}$ | $1.11_{\pm0.01}$ | $0.59_{\pm0.03}$ | $\mathbf{0.72_{\pm0.02}}$ | $\mathbf{0.02_{\pm0.58}}$ | $1.11_{\pm0.01}$ | $0.58_{\pm0.02}$ | $8.63_{\pm4.20}$ | $1.65_{\pm0.74}$ | $0.52_{\pm0.01}$ | $0.08_{\pm0.54}$ |
| TB | $1.09_{\pm0.02}$ | $0.51_{\pm0.04}$ | $0.76_{\pm0.02}$ | $0.48_{\pm0.04}$ | $1.09_{\pm0.02}$ | $0.47_{\pm0.10}$ | $0.74_{\pm0.01}$ | $0.45_{\pm0.03}$ | $1.07_{\pm0.01}$ | $0.42_{\pm0.11}$ | $10.86_{\pm5.22}$ | $2.29_{\pm1.35}$ | $0.54_{\pm0.01}$ | $0.26_{\pm0.06}$ |
| TB + LS | $1.46_{\pm0.02}$ | $0.66_{\pm0.03}$ | $1.13_{\pm0.03}$ | $0.40_{\pm0.02}$ | $1.40_{\pm0.09}$ | $0.62_{\pm0.08}$ | $1.11_{\pm0.18}$ | $0.46_{\pm0.09}$ | $1.41_{\pm0.02}$ | $0.62_{\pm0.07}$ | $268.47_{\pm327.21}$ | $29.70_{\pm46.66}$ | $1.01_{\pm0.03}$ | $0.36_{\pm0.04}$ |
| VarGrad | $1.09_{\pm0.02}$ | $0.50_{\pm0.05}$ | $0.76_{\pm0.02}$ | $0.42_{\pm0.05}$ | $1.11_{\pm0.01}$ | $0.36_{\pm0.24}$ | $0.76_{\pm0.01}$ | $0.46_{\pm0.06}$ | $1.07_{\pm0.02}$ | $0.46_{\pm0.04}$ | $9.97_{\pm4.49}$ | $2.41_{\pm1.20}$ | $0.53_{\pm0.01}$ | $0.17_{\pm0.18}$ |
| VarGrad + LS | $1.68_{\pm0.11}$ | $0.65_{\pm0.04}$ | $1.48_{\pm0.21}$ | $0.37_{\pm0.16}$ | $1.58_{\pm0.07}$ | $0.32_{\pm0.22}$ | $1.28_{\pm0.02}$ | $0.45_{\pm0.06}$ | $1.51_{\pm0.06}$ | $0.59_{\pm0.02}$ | $78.04_{\pm90.93}$ | $3.93_{\pm6.23}$ | $1.11_{\pm0.05}$ | $\mathbf{0.02_{\pm0.56}}$ |
| PIS + LP | $1.11_{\pm0.01}$ | $0.56_{\pm0.07}$ | $0.71_{\pm0.01}$ | $\mathbf{0.28_{\pm0.09}}$ | $1.10_{\pm0.01}$ | $0.56_{\pm0.04}$ | $\mathbf{0.69_{\pm0.02}}$ | $0.29_{\pm0.05}$ | $1.10_{\pm0.02}$ | $0.57_{\pm0.02}$ | $8.85_{\pm2.48}$ | $1.80_{\pm0.74}$ | $0.50_{\pm0.03}$ | $\mathbf{0.13_{\pm0.17}}$ |
| TB + LP | $1.08_{\pm0.02}$ | $0.40_{\pm0.12}$ | $0.72_{\pm0.03}$ | $0.37_{\pm0.03}$ | $1.54_{\pm0.51}$ | $0.50_{\pm0.12}$ | $0.91_{\pm0.21}$ | $0.44_{\pm0.11}$ | $1.07_{\pm0.02}$ | $0.38_{\pm0.11}$ | $30.07_{\pm22.61}$ | $9.56_{\pm13.27}$ | $0.48_{\pm0.005}$ | $0.25_{\pm0.03}$ |
| TB + LS + LP | $1.30_{\pm0.02}$ | $0.46_{\pm0.05}$ | $0.90_{\pm0.04}$ | $0.30_{\pm0.05}$ | $1.27_{\pm0.01}$ | $0.45_{\pm0.09}$ | $0.86_{\pm0.04}$ | $0.32_{\pm0.03}$ | $1.26_{\pm0.03}$ | $0.43_{\pm0.03}$ | $149.16_{\pm187.71}$ | $14.23_{\pm19.54}$ | $0.82_{\pm0.04}$ | $0.25_{\pm0.09}$ |
| VarGrad + LP | $1.08_{\pm0.02}$ | $0.46_{\pm0.17}$ | $0.72_{\pm0.02}$ | $0.37_{\pm0.02}$ | $1.10_{\pm0.01}$ | $0.43_{\pm0.08}$ | $0.74_{\pm0.02}$ | $0.38_{\pm0.04}$ | $1.07_{\pm0.01}$ | $0.43_{\pm0.13}$ | $48.10_{\pm42.22}$ | $21.80_{\pm30.37}$ | $0.48_{\pm0.01}$ | $0.23_{\pm0.04}$ |
| VarGrad + LS + LP | $1.39_{\pm0.04}$ | $0.46_{\pm0.04}$ | $0.99_{\pm0.05}$ | $0.33_{\pm0.03}$ | $1.44_{\pm0.04}$ | $0.44_{\pm0.08}$ | $1.09_{\pm0.18}$ | $0.36_{\pm0.06}$ | $1.32_{\pm0.05}$ | $0.44_{\pm0.04}$ | $162.54_{\pm189.06}$ | $10.68_{\pm12.41}$ | $0.77_{\pm0.07}$ | $0.25_{\pm0.05}$ |

### Manywell ($d = 32$)

| Training discretization → | 10-step random | | | | 10-step uniform | | | | 100-step uniform | |
|---|---|---|---|---|---|---|---|---|---|---|
| Evaluation steps → | 10 | | 100 | | 10 | | 100 | | 100 | |
| Algorithm ↓ Metric → | $\Delta \log Z$ | $\Delta \log Z^{\mathrm{RW}}$ | $\Delta \log Z$ | $\Delta \log Z^{\mathrm{RW}}$ | $\Delta \log Z$ | $\Delta \log Z^{\mathrm{RW}}$ | $\Delta \log Z$ | $\Delta \log Z^{\mathrm{RW}}$ | $\Delta \log Z$ | $\Delta \log Z^{\mathrm{RW}}$ |
| PIS ($\mathrm{lr} = 10^{-3}$) | $14.08_{\pm0.14}$ | $2.70_{\pm0.30}$ | $\mathbf{4.74_{\pm0.15}}$ | $2.77_{\pm0.05}$ | $14.08_{\pm0.13}$ | $2.97_{\pm0.37}$ | $69.72_{\pm13.41}$ | $33.84_{\pm11.79}$ | $\mathbf{3.87_{\pm0.03}}$ | $2.69_{\pm0.03}$ |
| PIS ($\mathrm{lr} = 10^{-4}$) | $14.34_{\pm0.28}$ | $3.23_{\pm0.54}$ | $6.37_{\pm0.08}$ | $2.80_{\pm0.20}$ | $14.16_{\pm0.27}$ | $2.86_{\pm0.73}$ | $75.30_{\pm1.89}$ | $35.65_{\pm1.45}$ | $4.17_{\pm0.04}$ | $2.62_{\pm0.06}$ |
| TB | $14.96_{\pm0.22}$ | $2.92_{\pm1.10}$ | $5.49_{\pm0.43}$ | $2.70_{\pm0.11}$ | $14.81_{\pm0.17}$ | $2.55_{\pm2.05}$ | $62.95_{\pm10.12}$ | $30.07_{\pm5.79}$ | $4.05_{\pm0.05}$ | $2.75_{\pm0.01}$ |
| TB + LS | $15.24_{\pm0.62}$ | $1.54_{\pm0.77}$ | $7.24_{\pm0.46}$ | $\mathbf{0.55_{\pm0.43}}$ | $14.86_{\pm0.60}$ | $0.45_{\pm0.89}$ | $51.08_{\pm4.27}$ | $16.82_{\pm3.08}$ | $4.52_{\pm0.91}$ | $\mathbf{0.37_{\pm0.14}}$ |
| VarGrad | $14.94_{\pm0.28}$ | $2.79_{\pm1.35}$ | $5.64_{\pm0.56}$ | $2.77_{\pm0.05}$ | $14.80_{\pm0.14}$ | $2.86_{\pm1.61}$ | $71.71_{\pm18.54}$ | $35.53_{\pm11.51}$ | $4.04_{\pm0.11}$ | $2.78_{\pm0.04}$ |
| VarGrad + LS | $16.02_{\pm0.26}$ | $2.84_{\pm0.15}$ | $7.03_{\pm0.56}$ | $2.00_{\pm0.46}$ | $16.08_{\pm0.75}$ | $3.26_{\pm1.10}$ | $69.14_{\pm12.35}$ | $28.45_{\pm13.46}$ | $6.53_{\pm3.56}$ | $4.43_{\pm2.70}$ |
| PIS + LP ($\mathrm{lr} = 10^{-3}$) | $13.97_{\pm0.18}$ | $2.15_{\pm0.28}$ | $4.34_{\pm0.25}$ | $1.69_{\pm0.41}$ | *diverging* | | | | $3.60_{\pm0.06}$ | $1.37_{\pm0.22}$ |
| PIS + LP ($\mathrm{lr} = 10^{-4}$) | $31.98_{\pm0.09}$ | $4.46_{\pm3.45}$ | $17.55_{\pm0.26}$ | $1.39_{\pm0.64}$ | $31.87_{\pm0.21}$ | $5.26_{\pm3.39}$ | $35.96_{\pm0.34}$ | $8.42_{\pm1.61}$ | $14.71_{\pm0.07}$ | $0.50_{\pm0.75}$ |
| TB + LP | $14.87_{\pm0.36}$ | $3.02_{\pm1.23}$ | $4.72_{\pm0.27}$ | $2.66_{\pm0.03}$ | $14.62_{\pm0.21}$ | $3.27_{\pm1.19}$ | $19.66_{\pm1.49}$ | $4.20_{\pm0.63}$ | $3.66_{\pm0.25}$ | $2.42_{\pm0.32}$ |
| TB + LS + LP | $13.88_{\pm0.58}$ | $0.60_{\pm0.23}$ | $\mathbf{2.40_{\pm0.39}}$ | $\mathbf{0.00_{\pm0.20}}$ | $13.67_{\pm0.44}$ | $0.81_{\pm0.51}$ | $24.32_{\pm1.02}$ | $2.10_{\pm0.43}$ | $1.81_{\pm0.05}$ | $\mathbf{0.03_{\pm0.07}}$ |
| VarGrad + LP | $14.79_{\pm0.39}$ | $3.11_{\pm1.11}$ | $4.68_{\pm0.34}$ | $2.71_{\pm0.03}$ | $14.63_{\pm0.20}$ | $3.15_{\pm0.02}$ | $20.72_{\pm3.32}$ | $3.89_{\pm0.72}$ | $3.41_{\pm0.10}$ | $2.09_{\pm0.27}$ |
| VarGrad + LS + LP | $16.24_{\pm0.70}$ | $1.31_{\pm0.75}$ | $5.12_{\pm0.68}$ | $0.32_{\pm0.21}$ | $14.22_{\pm0.22}$ | $0.35_{\pm0.08}$ | $22.89_{\pm4.12}$ | $1.71_{\pm1.87}$ | $\mathbf{1.77_{\pm0.06}}$ | $0.05_{\pm0.06}$ |

Table 3: ELBOs with different numbers of training and integration steps on **Credit**, **Cancer**, the conditional **VAE**, and **LGCP**. Training on **LGCP** was often unstable, consistent with findings of prior work, so fewer methods are reported.

**Credit** $(d = 25)$

| Training discretization → | 10-step random | | 10-step equidistant | | 10-step uniform | | 100-step uniform |
|---|---|---|---|---|---|---|---|
| Algorithm ↓ Evaluation steps → | 10 | 100 | 10 | 100 | 10 | 100 | 100 |
| PIS | $-1174.23_{\pm 14.07}$ | $-671.68_{\pm 8.14}$ | $-1181.62_{\pm 17.17}$ | $\mathbf{-667.03_{\pm 21.25}}$ | $-1171.35_{\pm 14.59}$ | $-1130.57_{\pm 20.69}$ | $\mathbf{-606.61_{\pm 0.65}}$ |
| TB | $-1301.50_{\pm 9.68}$ | $-911.04_{\pm 16.74}$ | $-1318.14_{\pm 22.13}$ | $-898.98_{\pm 24.18}$ | $-1281.31_{\pm 9.74}$ | $-1179.87_{\pm 30.61}$ | $-634.08_{\pm 2.88}$ |
| VarGrad | $-1279.95_{\pm 14.36}$ | $-847.65_{\pm 22.65}$ | $-1288.40_{\pm 10.49}$ | $-838.67_{\pm 14.12}$ | $-1264.02_{\pm 15.67}$ | $-1172.46_{\pm 32.20}$ | $-631.84_{\pm 3.20}$ |
| PIS + LP | $-1175.46_{\pm 14.14}$ | $-671.60_{\pm 12.01}$ | $-1183.60_{\pm 17.90}$ | $\mathbf{-669.30_{\pm 16.34}}$ | $-1174.25_{\pm 17.00}$ | $-1114.56_{\pm 43.56}$ | $-608.29_{\pm 2.12}$ |
| TB + LP | $-1342.96_{\pm 6.77}$ | $-943.63_{\pm 18.37}$ | $-1360.68_{\pm 32.84}$ | $-956.97_{\pm 4.12}$ | $-1300.17_{\pm 8.29}$ | $-1165.11_{\pm 25.76}$ | $-666.49_{\pm 2.79}$ |
| VarGrad + LP | $-1303.67_{\pm 15.11}$ | $-876.12_{\pm 10.70}$ | $-1323.16_{\pm 3.03}$ | $-933.40_{\pm 50.79}$ | $-1281.15_{\pm 6.49}$ | $-1186.95_{\pm 150.69}$ | $-651.98_{\pm 0.18}$ |

**Cancer** $(d = 31)$

| Training discretization → | 10-step random | | 10-step equidistant | | 10-step uniform | | 100-step uniform |
|---|---|---|---|---|---|---|---|
| Algorithm ↓ Evaluation steps → | 10 | 100 | 10 | 100 | 10 | 100 | 100 |
| PIS | $-6.60_{\pm 1.60}$ | $\mathbf{9.51_{\pm 3.13}}$ | $-7.73_{\pm 0.63}$ | $9.15_{\pm 1.45}$ | $-8.94_{\pm 4.87}$ | $-4933.64_{\pm 986.02}$ | $\mathbf{17.64_{\pm 12.51}}$ |
| TB | $-48.57_{\pm 23.39}$ | $-28.02_{\pm 18.77}$ | $-59.77_{\pm 45.25}$ | $-29.81_{\pm 18.81}$ | $-35.42_{\pm 8.76}$ | $-1096.80_{\pm 530.21}$ | $5.32_{\pm 6.03}$ |
| VarGrad | $-28.97_{\pm 6.03}$ | $-5.84_{\pm 0.98}$ | $-31.83_{\pm 2.58}$ | $-11.76_{\pm 5.90}$ | $-30.09_{\pm 3.76}$ | $-966.70_{\pm 357.24}$ | $9.41_{\pm 1.77}$ |
| PIS + LP | $-12.27_{\pm 2.99}$ | $\mathbf{7.30_{\pm 1.92}}$ | $-16.87_{\pm 3.26}$ | $6.35_{\pm 2.27}$ | $-11.51_{\pm 1.76}$ | $-3649.25_{\pm 629.76}$ | $\mathbf{19.47_{\pm 1.87}}$ |
| TB + LP | $-25.79_{\pm 3.04}$ | $-4.33_{\pm 2.77}$ | $-41.52_{\pm 28.79}$ | $-12.60_{\pm 16.39}$ | $-24.33_{\pm 1.48}$ | $-2738.75_{\pm 344.22}$ | $11.56_{\pm 0.59}$ |
| VarGrad + LP | $-30.55_{\pm 0.14}$ | $-1.69_{\pm 1.94}$ | $-28.16_{\pm 4.40}$ | $-6.05_{\pm 4.59}$ | $-26.36_{\pm 1.95}$ | $-978.60_{\pm 140.28}$ | $13.41_{\pm 2.19}$ |

**VAE** $(d = 20)$

| Training discretization → | 10-step random | | 10-step equidistant | | 10-step uniform | | 100-step uniform |
|---|---|---|---|---|---|---|---|
| Algorithm ↓ Evaluation steps → | 10 | 100 | 10 | 100 | 10 | 100 | 100 |
| PIS | $-117.83_{\pm 1.25}$ | $-104.52_{\pm 0.36}$ | $-117.68_{\pm 1.29}$ | $\mathbf{-104.29_{\pm 0.58}}$ | $-117.74_{\pm 1.12}$ | $-154.88_{\pm 6.51}$ | $\mathbf{-102.71_{\pm 0.52}}$ |
| TB | $-161.97_{\pm 1.26}$ | $-149.86_{\pm 4.93}$ | $-162.72_{\pm 4.85}$ | $-149.76_{\pm 0.75}$ | $-160.49_{\pm 0.56}$ | $-161.90_{\pm 5.63}$ | $-142.88_{\pm 5.14}$ |
| VarGrad | $-122.04_{\pm 1.62}$ | $-109.45_{\pm 1.40}$ | $-170.51_{\pm 4.78}$ | $-159.71_{\pm 7.46}$ | $-120.98_{\pm 0.96}$ | $-133.39_{\pm 4.98}$ | $-104.16_{\pm 0.67}$ |
| PIS + LP | $-115.90_{\pm 0.64}$ | $-100.20_{\pm 0.33}$ | $-115.81_{\pm 0.31}$ | $\mathbf{-100.13_{\pm 0.06}}$ | $-115.83_{\pm 0.82}$ | $-120.61_{\pm 1.41}$ | $-99.34_{\pm 0.40}$ |
| TB + LP | $-140.41_{\pm 2.18}$ | $-114.80_{\pm 1.07}$ | $-140.72_{\pm 1.10}$ | $-114.81_{\pm 1.39}$ | $-137.54_{\pm 2.51}$ | $-136.64_{\pm 2.96}$ | $-109.25_{\pm 1.68}$ |
| VarGrad + LP | $-118.52_{\pm 1.47}$ | $-102.24_{\pm 0.27}$ | $-138.51_{\pm 0.70}$ | $-113.49_{\pm 1.39}$ | $-117.35_{\pm 0.99}$ | $-122.22_{\pm 0.70}$ | $\mathbf{-99.01_{\pm 0.27}}$ |

**LGCP** $(d = 1600)$

| Training discretization → | 10-step random | | 10-step uniform | | 100-step uniform |
|---|---|---|---|---|---|
| Algorithm ↓ Evaluation steps → | 10 | 100 | 10 | 100 | 100 |
| PIS | $-1471.16_{\pm 6.83}$ | $\mathbf{-1467.85_{\pm 2.59}}$ | $-1471.49_{\pm 11.66}$ | $-1729.56_{\pm 103.09}$ | $\mathbf{-1465.14_{\pm 20.76}}$ |
| TB | $-1618.86_{\pm 3.01}$ | $-1617.35_{\pm 1.34}$ | $-1617.33_{\pm 6.54}$ | $-1666.37_{\pm 13.78}$ | $-1619.89_{\pm 6.56}$ |
| TB + LS | $-1878.87_{\pm 23.04}$ | $-1880.52_{\pm 13.07}$ | $-1877.13_{\pm 18.69}$ | $-1705.60_{\pm 36.86}$ | $-1891.62_{\pm 4.77}$ |
| PIS + LP | $343.46_{\pm 0.31}$ | $472.24_{\pm 0.68}$ | $343.18_{\pm 0.33}$ | $-211.79_{\pm 293.49}$ | $\mathbf{473.74_{\pm 1.14}}$ |
| TB + LP | $332.16_{\pm 0.42}$ | $461.53_{\pm 1.16}$ | $337.37_{\pm 0.12}$ | $-1931.42_{\pm 2636.38}$ | $468.68_{\pm 4.13}$ |
| TB + LS + LP | $341.53_{\pm 0.36}$ | $\mathbf{472.43_{\pm 0.42}}$ | $341.65_{\pm 0.16}$ | $-77.64_{\pm 77.72}$ | $451.89_{\pm 3.28}$ |

## D.2 Additional figures

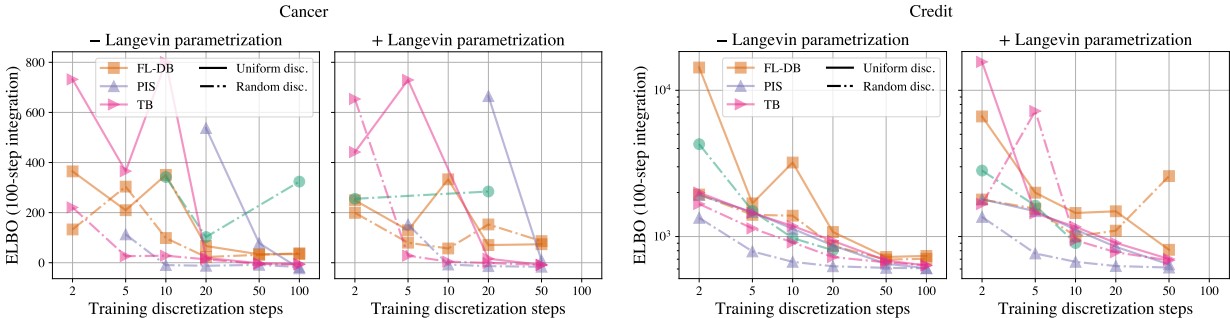

Figure 8: Results extending main text Fig. 5: Credit and Cancer densities.

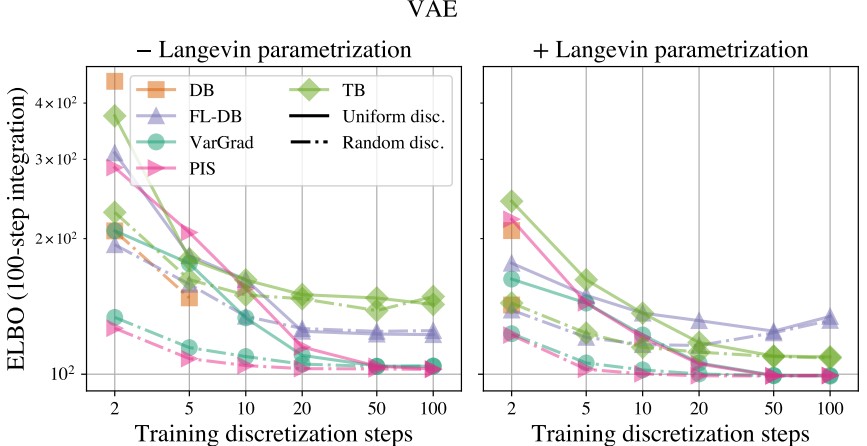

Figure 9: Results extending main text Fig. 5 on the conditional VAE target density.

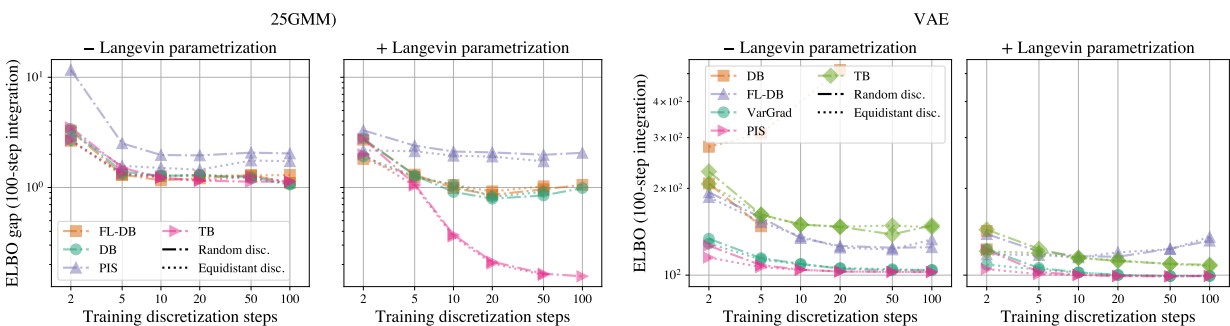

Figure 10: Comparison of **Random** and **Equidistant** distretizations on the **25GMM** (unconditional) and **VAE** (conditional) targets.

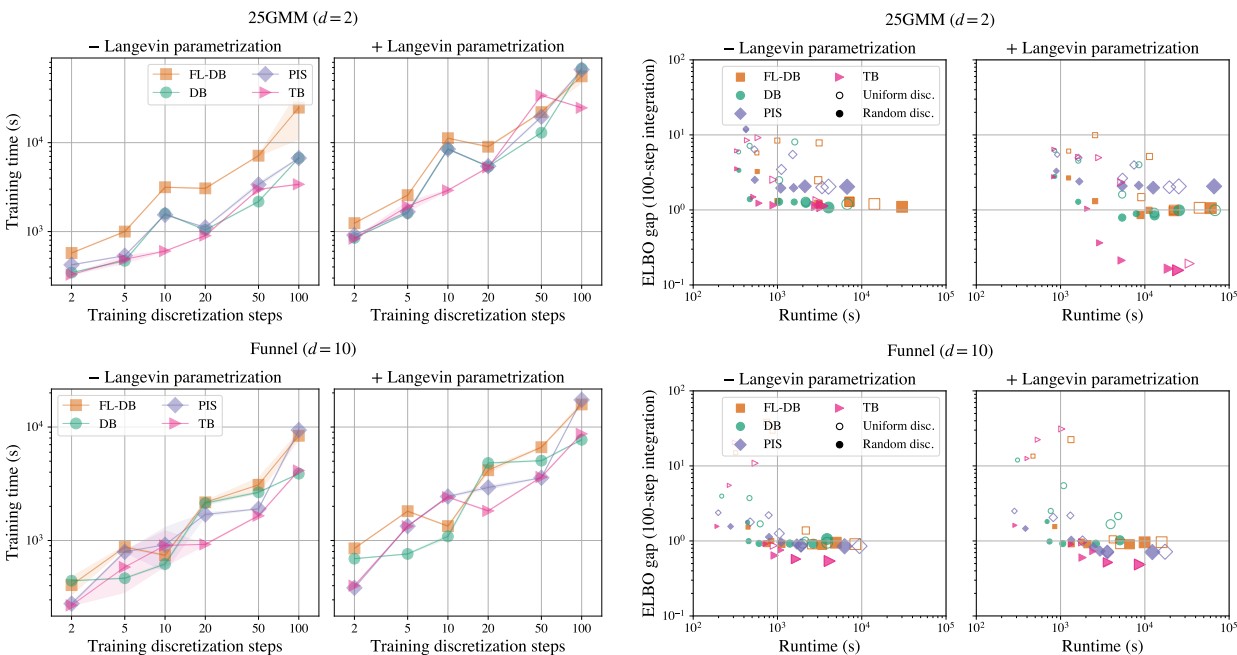

Figure 11: Results extending main text Fig. 6: Efficiency of nonuniform coarse discretizations on Funnel and 25GMM densities.

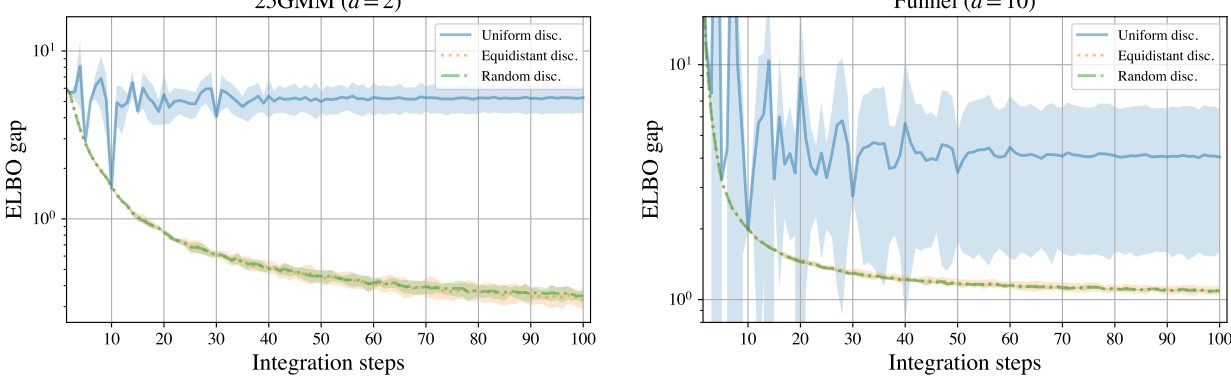

Figure 12: Results extending main text Fig. 7. Evaluation of models trained with $N_{\text{train}} = 10$ steps using varying numbers of integration steps.

### D.3 VAE reconstructions

MNIST data

VAE reconstruction (pretrained encoder)

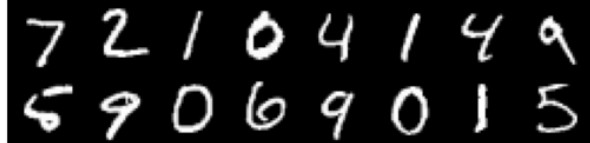
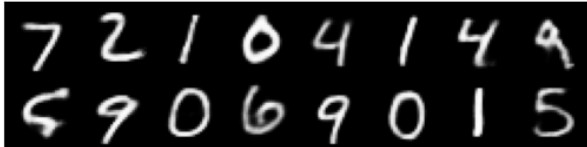

10-step **Uniform** training
Reconstruction from 100-step integration

10-step **Uniform** training
Reconstruction from 10-step integration

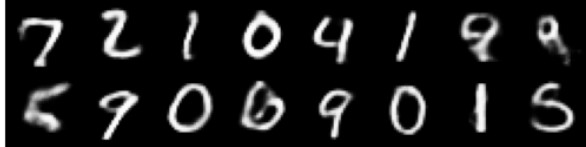
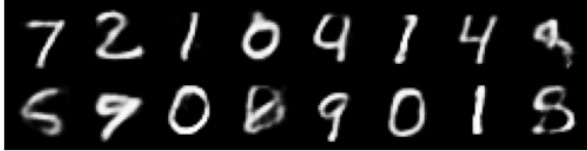

10-step **Random** training
Reconstruction from 100-step integration

10-step **Random** training
Reconstruction from 10-step integration

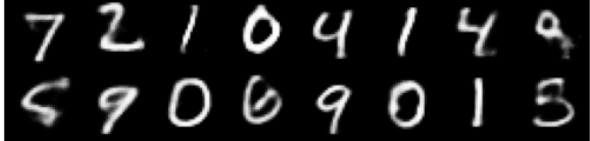
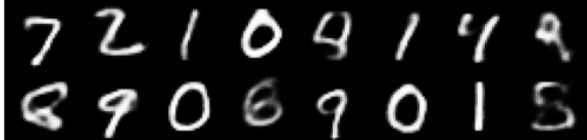

10-step **Equidistant** training
Reconstruction from 100-step integration

10-step **Equidistant** training
Reconstruction from 10-step integration

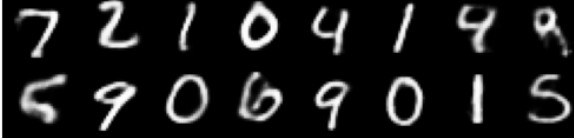
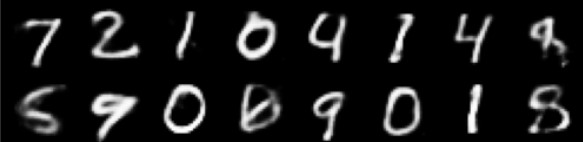

Figure 13: Mode of decoder $p(\cdot \mid z)$ evaluated on encoded latents $z$ for the VAE experiment: input data $x$ and reconstruction using $z$ sampled from the pretrained VAE encoder (top row) and reconstructions using $z$ sampled from diffusion encoders (next three rows).

