# OpenReview forum: "From discrete-time policies to continuous-time diffusion samplers: Asymptotic equivalences and faster training"
_TMLR — Accepted by TMLR_

### Review · Reviewer_Yf67 · 2025-07-21

**Summary Of Contributions:**

This paper studies diffusion samplers, which have met great empirical success in various tasks pertaining to the field of generative AI. The main focus is on the connection between discrete-time approaches, which are used in practice (e.g., when combined with RL methods), and continuous-time models involving stochastic differential equations. The authors focus on two types of objectives: global in time and local in time. They show, under suitable regularity assumptions, that in both cases the discrete-time objectives converge to their continuous-time analogues. Numerical experiments on several sampling problems from the literature illustrate the advantages of using nonuniform time discretizations during the training of diffusion models.

**Audience:**

Yes

**Broader Impact Concerns:**

Nothing specific.

**Claims And Evidence:**

Yes

**Requested Changes:**

Here are a few points which need further clarification in the paper:

1. Page 3, “can allow for greatly improved sample efficiency”: With respect to what baseline is the improvement supposed to be?

2. Page 3, point (3): Could you please clarify how this is supported by Fig. 5 (and Fig. 11)? Here it is written “with nonuniform time discretizations” but it seems a bit misleading because in the experiments, you use not just “nonuniform” but more precisely *random* discretizations with a specific distribution, if I understand correctly. I suspect this is important because one could probably construct “nonuniform time discretizations” leading to very bad performance (e.g., if we put almost all the points close to time 0 and then just a big time interval until terminal time). Also, what do you mean by “similar performance to state-of-the-art methods”? Do you simply mean “similar performance as with uniform meshes”?

3. Page 3, “The results in §3 extend classical results on SDE approximations to objectives for diffusion-based samplers”: Could you please be more specific? The literature on discretization of SDEs is very broad and it is, a priori, surprising that no existing result could be applied. On pages 2 and 3, only the book of Kloeden and Platen is cited, but there are many other results. It would be good to clarify which aspect(s) of the problem make it impossible to apply existing results.

4. Page 3, “However, it is close to the setting of maximum-entropy reinforcement learning”: I would recommend writing clearly the control problem in a mathematical formulation (you write “We relegate formal definitions to Appendix B.2” but actually the problem is not really defined precisely even in the appendix), and comparing rigorously the definition with the more standard RL framework. If we include an entropy term in the reward, do we recover exactly your problem or only approximately?

5. Page 6, equation (13): What is the definition of $dX_t$ in this situation? Is the stochastic integral well-defined or is it just an informal derivation? Similarly, could you please clarify the meaning of the last two integrals in (14)?

6. Page 6, here and elsewhere in the paper, many references are given but not precisely enough for the reader to efficiently extract information. For example “A related result was derived by Richter & Berner (2024)”: Which result? Another example: on page 7, “The result can, e.g., be found in Kloeden & Platen (1992)”. More generally, when citing references for specific results or formulas, please quote the corresponding Theorem/Proposition number or equation number from that reference.

7. Page 7, “more relevant for practical applications”: This is perhaps explained somewhere else in the paper, but could you please clarify what such results bring from the application viewpoint? (This is not a criticism, simply an honest question.)

8. Page 8, “The convergence holds for the TB divergence with respect to any $c$”: What do you mean by “with respect to”? Is this comment simply saying that you can apply Proposition 3.3 with $f(x) = (x-c)^2$, or is there something more?

9. Page 10, “While recent work on diffusion samplers has used a discretization with uniform-length time intervals for both integration and training, we vary the time discretization.”: It seems that the impact of using nonuniform time intervals is not discussed in the theoretical analysis of Section 3. For example, Propositions 3.2 and 3.4 only make an assumption on the largest time step going to 0, but they do not distinguish between uniform and nonuniform meshes. Is it possible to analyze theoretically the differences between such meshes to support your numerical findings?

10. Page 19, Appendix B.1 (Assumptions): Could you please provide sufficient conditions ensuring that “all SDEs admit densities of their time marginals (w.r.t. the Lebesgue measure) that are sufficiently smooth such that we have strong solutions to the corresponding Fokker-Planck equations”? This condition is quite abstract and it is hard for the reader to check whether it holds for their model. Also, what condition do we need on the neural network such that these assumptions are satisfied when using a neural network for the drift in the SDE (as you mention below (9))?

**Strengths And Weaknesses:**

**Strengths:**
- Generally speaking, the paper is well written.
- The first two sections present the background information in a synthetic and well-organized manner.
- The convergence results, although based on existing results for Euler-Maruyama schemes, are nicely tailored to the objectives used in diffusion samplers.
- The numerical results provide information on aspects such as rate of convergence and sensitivity to different types of meshes.
- The numerical experiments cover various examples from the literature and are nicely presented.

**Weaknesses:**
- It would be helpful for the reader if the paper were more specific when citing references; otherwise, it is sometimes quite difficult to recover what the authors had in mind.
- The theoretical results seem to be of limited interest for applications because they do not provide any insights in terms of rate of convergence or sensitivity to different types of meshes.

---

> ### Author Response · Authors · 2025-11-13
>
> We thank the reviewer for their positive and thorough feedback. We are glad they found the paper "well written" and the background "well-organized," and that they appreciated how our theoretical results are "nicely tailored" to diffusion sampler objectives and are supported by "nicely presented" experiments.
>
> We will revise the paper to incorporate the following clarifications, which address the reviewer's points.
>
> ### On citations
> > "It would be helpful... if the paper were more specific when citing references... please quote the corresponding Theorem/Proposition number or equation number."
>
> Thank you for raising this point. We will improve the manuscript by making our citations more precise and update key citations with specific Theorem, Proposition, or Equation numbers, making it easier for the reader to trace our claims.
>
> ### On the practical utility of the theory
>
> > "The theoretical results seem to be of limited interest for applications because they do not provide any insights in terms of rate of convergence or sensitivity to different types of meshes... Is it possible to analyze theoretically the differences between such meshes?"
>
> Thank you for making this point. Our most important contribution is showing consistency of objectives, rather than giving practical numerical bounds. We do obtain the order of convergence for the DB objective. Moreover, based on our findings -- where equally spaced but shifted intermediate time points during training work about equally well as random ones -- we can conclude that the reason for training with uniform discretisation working badly is not to do with the rate of convergence to a continuous-time objective, but rather with OOD values of the time input $t$ that we get at inference time if only a small, fixed set of values of $t$ is seen during training.
>
> (continued below)

---

> > ### Author Response · Authors · 2025-11-13
> >
> > (continued from above)
> >
> > ### Responses to specific requested changes
> >
> > Thank you for raising these issues. We respond to each of them in the following and will appropriate address them in the final version.
> >
> > > **Page 3, "can allow for greatly improved sample efficiency": With respect to what baseline?**
> >
> > The improvement is relative to the *standard practice* of training with a fine, uniform time discretization (e.g., $N_{train} = 100$) and we will edit this sentence to be more explicit.
> >
> > > **Page 3, point (3): Clarify "nonuniform" vs. "random" and "similar performance to state-of-the-art".**
> >
> > This is a great point. "Nonuniform" is the general category, while "Random" and "Equidistant" are the specific schemes we test. The reviewer is correct that a *bad* non-uniform mesh could be constructed. By "similar performance to state-of-the-art methods," we mean "similar performance to the same objectives trained with a fine, uniform mesh (e.g., $N_{train}=100$), which is the common practice."
> >
> > > **Page 3, "extend classical results on SDE approximations": Please be more specific.**
> >
> > Our contribution is not in re-proving SDE convergence. It is in *tailoring* these results to the specific, complex *training objectives* used by modern samplers. Classical results (like in Kloeden & Platen) typically show convergence of the *process* $\hat{X}$ to $X$ or of $\mathbb{E}[f(\hat{X}_T)]$. Our Prop 3.2 and 3.4 are novel in that they show convergence for the *losses themselves*—which are functionals of the *entire path* (like the TB/VarGrad losses) or specific log-ratios (the DB loss)—to their continuous-time analogues (the path KL divergence and the FPE, respectively).
> >
> > > **Page 3, "maximum-entropy reinforcement learning": Clarify the control problem.**
> >
> > Please see the beginning of the response to Reviewer RDmC; we will add extended discussion of this connection to the paper.
> >
> > > **Page 6, equation (13) & (14): Clarify $\vec{\mu}^{(2)}$ and the integrals.**
> >
> > $\vec{\mu}^{(1)}$ and $\vec{\mu}^{(2)}$ are the drift functions for the two SDEs defining path measures $\mathbb{P}^{(1)}$ and $\mathbb{P}^{(2)}$. The integrals w.r.t. $dX_t$ and $d\bar{X}_t$ are well-defined Itô integrals and the result follows from Girsanov's theorem. We will add a precise citation for Girsanov's theorem.
> >
> > > **Page 7, "more relevant for practical applications": How do the theoretical results help?**
> >
> > First of all, "more relevant for practical applications" at the start of this section means simply that while the previous section (§3.1) showed the convergence of path space measures, the present one (§3.2) studies the convergence of divergences between path space measures that can be used as training objectives.
> >
> > Consistency of the integration scheme alone would not tell us anything about the divergences we use to match processes. However, the results about divergences do have implications for training. Specifically, they show the validity of training in one time discretisation (or multiple) and sampling in another. Without the consistency results, there would be no reason to think that objectives with, for example, 50 and 100 discretisation steps would not give competing learning signals. However, we show that in the limit of fine discretisation, all objectives converge to the same objective and would have the same minimiser. This justifies the practice of training with coarse discretisations to save computation.
> >
> > > **Page 8, "The convergence holds for the TB divergence with respect to any $c$": What do you mean?**
> >
> > This refers to the common practice of subtracting a baseline $c$ or a learned log-partition function $\log \hat{Z}$ from the log-ratio before squaring it, i.e., $D_{TB} = \mathbb{E}[( \log \frac{d\hat{\mathbb{P}}}{d\hat{\mathbb{Q}}} - c)^2]$. Our point is that Prop. 3.3, which uses $f(x)=x^2$, also holds for $f(x)=(x-c)^2$, so our convergence result is robust to this practical variant. We will rephrase this in the final version.
> >
> > > **Page 19, Appendix B.1 (Assumptions): Provide sufficient conditions.**
> >
> > Continuous differentiability and polynomial growth in $x_t$ of the SDE drifts, uniformly in $t$, should be sufficient (but we will specify a simple set of sufficient conditions in the final version). In general, neural networks of typical architectures (such as MLPs) with infinitely differentiable activations would satisfy such a constraint.

---

### Review · Reviewer_RDmC · 2025-07-28

**Summary Of Contributions:**

The paper proposes a theoretical approach to using continuous time sampling techniques for discretized training in practice. The paper derives the theoretical approach, guarantee and algorithm for adapting discretized training to continuous sampling, hence accelerating the sampling procedure significantly. The paper shows improvements over baseline methods for speed and quality)

**Audience:**

Yes

**Claims And Evidence:**

Yes

**Requested Changes:**

===== *applications to rl cases* =====

If I understand the methods correctly, the fast sampling comes from the fact that when sampling from the model itself we can leverage continuous time techniques and hence the speed up comes from sampling from a parameterized model itself. If we want to apply the method to the general RL case, where we alternate between sampling from models and the environment, the application becomes more challenging. Is it fair to say so? I think it'd be nice to discuss the implication of such cases because as the title suggests "policies", I'd assume it implies applications to RL specific use cases, but I think the applications in the paper are pure generative modeling.

===== *figure 5 right plots* =====

The figure 5 right plot shows that random discretization provides a good balance between speed and quality, as shown on the parato fronts. However, the comparison between different training methods is a bit unclear to me. Here, I assume is testing for inference time performance and so the model has finished training, hence we are indirectly comparing the quality of base models trained by different methods? Is that correct?

Since we integrate over 100-step, as indicated on the y-axis annotations, how do we trace out a curve for each specific method, i.e., what's the variation here between e.g., the pink triangle dots with different sizes? As a nit point, it'd be nice to visualize better so as to be able to compare different methods' parato fronts in a more informative way.

===== *figure 4 right plots* =====

For the TB and FL-DB methods, why is there a clear increase in the ELBO gap as the number of discretization step increases from 2 to <10 in the plot? Intuitively the quality of approximation should improve as the discretization step increases, and here it seems that the results are suggestive of noise or something else going up that's not captured by the intuition. Can the authors provide some elaboration on this?

===== *higher d benchmarks* =====

It seems that the manywell test case for whcih $d=32$ is the highest dimensional test case that the paper investigates. Would it be possible to benchmark in scenarios where the dimensions are even higher? It would also be nice to plot the level of speed up against $d$, so as to better understand the scaling behavior of different methods as a function of the dimension $d$. After all it is of practical interest to handle extremely high dimensional inputs, and understanding trend even for moderate sized $d$ is a great step forward.

**Strengths And Weaknesses:**

The paper is interesting to machine learning community that can leverage similar continuous time techniques to speed up discretized training. I think the paper is quite solid in the theoretical and algorithmic fronts but maybe the empirical ablations can be made more solid and convincing as a way to improve the paper.

---

> ### Author Response · Authors · 2025-11-13
>
> Thank you for your positive feedback and for describing our work as "quite solid in the theoretical and algorithmic fronts". We appreciate your insightful questions, which will help us improve the clarity and impact of the paper.
>
> Below, we address each of your requested changes.
>
> ### Applications to RL
>
> > "I'd assume it implies applications to RL specific use cases, but I think the applications in the paper are pure generative modeling."
>
> The "policies" in the title do not suggest RL applications, but rather that certain families of algorithms for training diffusion samplers that we study are connected with objectives from entropy-regularised RL.
>
> The DB objective enforces the soft Bellman equation [1,2] in a certain MDP, and TB/LV are path consistency objectives [3], resulting from telescoping cancellation when summing the soft Bellman constraints along a trajectory. Thus training a sequential sampler using one of these off-policy objectives is the same as training a policy to solve the entropy-regularised reward maximisation problem, justifying the view of the sampler as a "policy".
>
> The fundamentals of this connection are most thoroughly described (in the discrete case) in [4,5], but we summarise them here. In short, one can define a deterministic MDP in which the states are possible values of the discrete-time latent $\hat X_n$ and actions pass from $\hat X_n$ to values of the next latent $\hat X_{n+1}$. A policy $\overrightarrow\pi$ in this MDP is then a family of conditional distributions defining a discrete-time process, as in Fig. 3 of the paper. We then define rewards in the following way for a fixed backward kernel $\overleftarrow\pi$:
> $$\begin{align*}
> r(\hat X_n,\hat X_{n+1})&=\log\overleftarrow\pi(\hat X_n\mid X_{n+1})\\
> r(\hat X_N,\bot)&=-\mathcal{E}(\hat X_N).
> \end{align*}$$
> The reward accumulated along a trajectory $(\hat X_0,\dots,\hat X_N,\bot)$ is then $-\mathcal{E}(\hat X_N)+\hat{\mathbb{P}}(\hat X\mid\hat X_N)$.
> It is known that solving the entropy-regularised RL problem $\max_{\overrightarrow\pi}\mathbb{E}_{\tau\sim\overrightarrow\pi}r(\tau)+\alpha\mathcal{H}(\pi)$ results in sampling trajectories $\tau$ from the Boltzmann distribution with density proportional to $\exp(-r(\tau)/\alpha)$. Setting $\alpha=1$, this achieves time reversal.
>
> [1] Ziebart et al., "Modeling purposeful adaptive behavior...", 2010.
>
> [2] Haarnoja et al., "Soft actor-critic...", 2018.
>
> [3] Nachum et al., "Bridging the gap between value and policy based reinforcement learning", 2017.
>
> [4] Tiapkin et al., "Generative flow networks as entropy-regularized RL", 2024.
>
> [5] Deleu et al., "Discrete probabilistic inference as control...", 2024.
>
> (continued below)

---

> > ### Author Response · Authors · 2025-11-13
> >
> > (continued from above)
> >
> >
> > ### Figure 5 (Pareto fronts)
> >
> > Yes, on these plots we compare the runtime (we could equally have plotted the time per training iteration) to the inference-time performance, where the latter always uses fixed 100-step integration.
> >
> > We neglected to write in the caption of Fig. 5 that the size of the marker is determined by the number of integration steps (dilation proportional to its square root, to be precise; we should expect the runtime to be approximately linear in is number).
> >
> > So, for example, one can consider the front formed by the solid pink triangles (TB, random discretisation) and the empty pink triangles (TB, uniform discretisation). Each solid triangle is approximately on the same vertical line as a corrsponding empty one, since the runtime does not depend on the discretisation. The larger instances of these shapes (near the right) are close to each other, showing that when the discretisation is fine, the choice of discretisation has little impact on the metric. However, the frontier formed by the solid markers is lower than that formed by empty markers, which indicates that the performance degrades less as the discretisation is coarsened when the time discretisation is random.
> >
> > ### Figure 4 plots (non-monotonic ELBO gap)
> >
> > > "why is there a clear increase in the ELBO gap as the number of discretization step increases from 2 to <10...?"
> >
> > Thanks for this observation. The intuition that "quality should improve as discretization steps increase" is correct for *inference-time integration* (as we show in Figure 6).
> >
> > However, *training* is a more complex optimization problem. The non-monotonic behavior you spotted in the very coarse regime (e.g., $N_{train}$ from 2 to 10) is likely due to the changing optimization landscape, that can lead to poor local minima.
> >
> >
> > ### Higher-dimensional benchmarks
> >
> > > "It seems that the manywell test case... is the highest dimensional test case... Would it be possible to benchmark in scenarios where the dimensions are even higher?"
> >
> > Thank you for this suggestion. We agree that scaling is critical. We would like to clarify that our study *does* include benchmarks with dimensions higher than Manywell ($d=32$). We test on:
> > * *VAE* ($d=20$)
> > * *Credit* ($d=25$)
> > * *Cancer* ($d=31$)
> > * *LGCP* ($d=1600$)
> >
> > In paritcular, the *Log-Gaussian Cox Process* (LGCP) is a 1600-dimensional problem and we report results for it in Table 3 (Appendix D.1). As noted in the appendix, training on this target was often unstable, which is consistent with the extreme challenge of high-dimensional sampling observed in other works.
> >
> > Nonetheless, our key findings held: on this 1600-dimensional task, models trained with coarse, non-uniform steps (10-step random) performed competitively with, and trained much faster than, those trained with 100 uniform steps.

---

### Review · Reviewer_srtk · 2025-10-02

**Summary Of Contributions:**

**Summary**

The paper develops a unifying view of several discrete-time policy learning objectives (Trajectory Balance, Detailed/Forward-Looking Detailed Balance, and a variance-reduced objective) by connecting them to continuous-time diffusion samplers via asymptotic analysis. In the small-stepsize limit, the authors show these discrete objectives converge to path-space divergences associated with continuous-time samplers, and they propose a general “policy-induced sampler” (PIS) objective that subsumes common objectives and suggests new ones. They then exploit this lens to argue—and demonstrate empirically—that training with coarser and non-uniform time discretizations can be substantially more efficient without sacrificing evaluation quality, especially when evaluation is performed with many integration steps.

**Contributions**

1. Asymptotic equivalences between discrete-time objectives (TB/DB/FL-DB/VarGrad) and continuous-time path-divergences, yielding a principled bridge between policy-learning and diffusion sampling.
2. PIS objective family that recovers TB, DB/FL-DB and VarGrad as special cases and provides a single training handle across tasks.
3. Efficiency via coarse & non-uniform training discretization. The paper compares uniform vs. equidistant/random (non-uniform) grids and finds that training on a coarse non-uniform grid and evaluating with a fine grid often yields the same (or better) ELBO/BO metrics at lower training time.
4. Empirical validation across synthetic targets (25GMM, Funnel), conditional VAE targets, and higher-dimensional settings (e.g., LGCP). Tables/figures report ELBO/BO gaps and wall-clock trends under different discretizations and algorithmic choices (including a “Langevin parametrization (LP)” variant).

**Audience:**

Yes

**Broader Impact Concerns:**

No concerns as far as I know.

**Claims And Evidence:**

Yes

**Requested Changes:**

Please see the weakness part.

**Strengths And Weaknesses:**

**Strengths**

* Conceptual unification. The asymptotic view clarifies how seemingly different discrete objectives relate to diffusion-like path measures; this helps practitioners reason about when an objective is appropriate and suggests principled hybrids (e.g., PIS).
* Actionable guidance. The finding that training can safely use coarse/non-uniform discretizations while evaluation uses fine grids is practical and easy to adopt; the paper substantiates this point with careful ablations across datasets and objectives.
* Breadth of objectives and targets. The work does not champion a single loss; instead it positions TB/DB/VarGrad/FL-DB/PIS within one family and evaluates them side-by-side on synthetic and conditional targets, including higher-D LGCP.


**Weaknesses**

* *Theoretical assumptions.*
The asymptotic equivalences rest on a vanishing step-size limit. In practice, training uses 2–20 steps while evaluation may use 100+ steps. It would help to quantify finite-stepsize error (e.g., via non-asymptotic bounds or empirical convergence curves versus step size) to indicate where the theory is a good proxy for practice. (The figures provide qualitative trends, but a bound or controlled ablation would strengthen the claim.)
* *Clarity on “non-uniform” schemes.*
The paper mentions “random” and “equidistant” discretizations; it would help to explicitly define the sampling procedures and any bias-variance trade-offs they introduce (e.g., are random grids re-drawn per minibatch? fixed over training? how are early vs late intervals weighted?). The captions point to these setups, but a short algorithm box would remove ambiguity.
* *Metrics and downstream impact.*
Most results use ELBO/BO gaps and reconstructions. Are there downstream sampling-quality metrics (e.g., for image VAEs) or lower-variance estimators of log-likelihood that better capture the benefit of the path-space view? The tables for VAE/Cancer are helpful, but an end-task comparison (or a runtime-vs-likelihood Pareto plot) would make the efficiency story better.
* *Generality beyond studied targets.*
The targets are well-chosen but still modest (e.g., 25GMM/Funnel/VAEs). A short discussion about extending to high-dimensional image or diffusion-policy tasks (and any stability issues anticipated there) would broaden the paper’s impact narrative.

---

> ### Author Response · Authors · 2025-11-13
>
> We thank the reviewer for their insightful and constructive feedback on our paper. We are pleased they recognized the value of our work's conceptual unification, its actionable guidance on training, and the breadth of objectives and targets studied. We will incorporate the following changes to address the points raised in the "Weaknesses" section.
>
> ### Clarification on objectives (PIS)
>
> First, we'd like to clarify a small point in the review summary regarding the "policy-induced sampler" (PIS). In our work, PIS refers to the *Path Integral Sampler* from Zhang & Chen (2022), which we study as one of the existing trajectory-level objectives alongside Trajectory Balance (TB). Our primary contribution is not proposing a new subsuming objective, but rather providing the unifying asymptotic analysis that connects discrete-time objectives (TB, Detailed Balance, and others) to their continuous-time counterparts. We will make this distinction clearer in the introduction.
>
> ### On the finite-stepsize error
>
> > "It would help to quantify finite-stepsize error... to indicate where the theory is a good proxy for practice...(The figures provide qualitative trends, but a bound or controlled ablation would strengthen the claim.)"
>
> While it is challenging to derive non-vacuous bounds, we designed the experiments in Figures 4, 8, and 9 to provide the controlled ablations of the "empirical convergence curves versus step size" the reviewer requested. These plots show the ELBO gap as a function of $N_{train}$ (from 2 to 100 steps) with all other parameters kept the same. Similarly, Figure 6 and Figure 12 show how models trained with a coarse, finite $N_{train}=10$ perform when evaluated at different *inference* step counts ($N_{eval}$).
>
> ### Definitions of "non-uniform" schemes
>
> > "it would help to explicitly define the sampling procedures [for 'random' and 'equidistant']... a short algorithm box would remove ambiguity."
>
> Thanks for this suggestion. These definitions are currently in the appendix, but we agree they are important enough for the main text. We will move the definition and illustrations of the *"Random"* and *"Equidistant"* schemes from Appendix C.2 into Section 4.
>
> ### Metrics and downstream impact
>
> > "an end-task comparison (or a runtime-vs-likelihood Pareto plot) would make the efficiency story better."
>
> We are unsure how to understand the suggestion, as runtime-ELBO Pareto fronts for many densities are shown in Figs. 5 and 11. We could add similar plots for VAE (they would tell a similar story). See the response to Reviewer RDmC (point 2) regarding interpretation of these figures.
>
> For the VAE experiment, we also provide qualitative, downstream *image reconstructions* presented in Appendix D.3 (Fig. 13) complementing the quantitative ELBO metrics. We note that the models trained with coarse, non-uniform steps produce reconstructions that are visually competitive with those from 100-step uniform training, reinforcing the efficiency-performance trade-off.
>
> ### Discussion on high-dimensional targets
>
> > "A short discussion about extending to high-dimensional image or diffusion-policy tasks... would broaden the paper’s impact narrative."
>
> This is an excellent point. We did include the 1600-dimensional *LGCP* task (results in Table 3) as a step in this direction, but we will expand our discussion in the Conclusion (Section 5). In particular, we believe that our findings are *especially* relevant for such high-dimensional tasks, where training with long trajectories is often prohibitively expensive. The success of *coarse, non-uniform steps* suggests a path toward such high-dimensional settings where memory and computation are primary bottlenecks.

---

### Decision · Action_Editor_1vLR · 2025-12-19

**Recommendation:** Accept as is

**Audience:**

Yes

**Audience Explanation:**

See above

**Claims And Evidence:**

Yes

**Claims Explanation:**

On the methodology and theory front, the paper is strong. The claims regarding the links between discrete and continuous versions of the objectives for the Path integral sampler (PIS), Trajectory Balance (TB) (and VarGrad) and the Detailed Balance loss (DB) from the GFlowNet literature. One of the reviewer found the paper well-written and I agree that it is a nice introduction to the field of training neural stochastic differential equations, or diffusion models, to sample from a Boltzmann distribution.

While the experimental part was pointed out as a weak part of the paper, the authors have addressed the technical concerns of the reviewers in their rebuttal.